# Hexosamine biosynthesis disruption impairs GPI production and arrests *Plasmodium falciparum* growth at schizont stages

María Pía Alberione[1,2], Yunuen Avalos-Padilla[1,3], Gabriel W. Rangel[4], Miriam Ramírez[1,2], Tais Romero-Uruñuela[1,2], Àngel Fenollar[1,2], Jonathan Ortega-Barrionuevo[1,2], Marcell Crispim[1,2,5,6], Terry K. Smith[7], Manuel Llinás[4,8], Luis Izquierdo[1,2,9]*

**1** ISGlobal, Barcelona, Spain, **2** Facultat de Medicina i Ciències de la Salut, Universitat de Barcelona (UB), Barcelona, Spain, **3** Nanomalaria Group, Institute for Bioengineering of Catalonia (IBEC), The Barcelona Institute of Science and Technology, Barcelona, Spain, **4** Department of Biochemistry and Molecular Biology and Huck Center for Malaria Research, Pennsylvania State University, University Park, Pennsylvania, United States of America, **5** Department of Parasitology, Institute of Biomedical Sciences of the University of São Paulo, São Paulo, Brazil, **6** Department of Clinical and Toxicological Analyses, School of Pharmaceutical Sciences, Federal University of Alfenas, Alfenas, Brazil, **7** Biomedical Sciences Research Complex (BSRC), School of Biology, University of St Andrews, St Andrews, United Kingdom, **8** Department of Chemistry, Pennsylvania State University, University Park, Pennsylvania, United States of America, **9** CIBER de Enfermedades Infecciosas (CIBERINFEC), Barcelona, Spain

* luis.izquierdo@isglobal.org

## Abstract

UDP-N-acetylglucosamine (UDP-GlcNAc) is a crucial sugar nucleotide for glycan synthesis in eukaryotes. In the malaria parasite *Plasmodium falciparum*, UDP-GlcNAc is synthesized via the hexosamine biosynthetic pathway (HBP) and is essential for glycosylphosphatidylinositol (GPI) anchor production, the most prominent form of protein glycosylation in the parasite. In this study, we explore a conditional knockout of glucosamine-6-phosphate N-acetyltransferase (*Pf*GNA1), a key HBP enzyme. *Pf*GNA1 depletion led to significant disruptions in HBP metabolites, impairing GPI biosynthesis and causing mislocalization of the merozoite surface protein 1 (MSP1), the most abundant GPI-anchored protein in the parasite. Furthermore, parasites were arrested at the schizont stage, exhibiting severe segmentation defects and an incomplete rupture of the parasitophorous vacuole membrane (PVM), preventing egress from host red blood cells. Our findings demonstrate the critical role of HBP and GPI biosynthesis in *P. falciparum* asexual blood stage development and underscore the potential of targeting these pathways as a therapeutic strategy against malaria.

### Author summary

Malaria remains a major cause of illness and death, particularly in sub-Saharan Africa, with increasing resistance to treatments highlighting the urgent need for

**Data availability statement:** The raw metabolomics data supporting the findings of this study have been deposited in the NIH Metabolomics Workbench under Study ID ST003642 and are publicly available at: https://www.metabolomicsworkbench.org. The dataset can be accessed via DOI: 10.21228/M8J242.

**Funding:** This research is part of the ISGlobal Program on the Molecular Mechanisms of Malaria which is partially supported by the Fundación Ramón Areces. We acknowledge support from the grant CEX2023-0001290-S funded by MCIN/AEI/ 10.13039/501100011033, and support from the Generalitat de Catalunya through the CERCA Program. LI received support by PID2022-137031OB-I00 grant funded by MCIN/AEI/10.13039/501100011033/ FEDER, UE. The work is supported by a FI Fellowship from the Generalitat de Catalunya supported by Secretaria d'Universitats i Recerca de la Generalitat de Catalunya and Fons Social Europeu (2021 FI_B 00470) and an EMBO Scientific Exchange Grant (9474) to MPA. GWR was supported by a Penn State Eberly Research Fellowship and grant R21AI174085 from the NIH. Funding to ML was supported by the Eberly College of Science and the Huck Institutes of the Life Sciences at The Pennsylvania State University and NIH grant R21AI174085. The funders had no role in study design, data collection and analysis, decision to publish, or preparation of the manuscript.

**Competing interests:** The authors have declared that no competing interests exist.

new strategies. Malaria parasites rely on the hexosamine biosynthetic pathway to produce UDP-N-acetylglucosamine, an essential metabolite for glycosylphosphatidylinositol synthesis. Glycosylphosphatidylinositol molecules anchor vital proteins to the parasite's surface and, as free glycolipids, serve as structural components of its membranes. Our study examined the effects of disrupting *Pf*GNA1, a key enzyme in the hexosamine biosynthetic pathway, which is distinct from its human counterparts. Disruption of *Pf*GNA1 blocked the production of glycosylphosphatidylinositol, leading to improper protein localization, developmental arrest, and failure of the parasites to mature or exit infected red blood cells. Our results underscore the central role of the hexosamine biosynthetic pathway and glycosylphosphatidylinositol biosynthesis, which are essential for parasite survival. This pathway represents a promising target for developing novel antimalarial therapies.

## Introduction

Malaria is a major global health problem that kills more than 600,000 people a year, mainly children under the age of five and pregnant women in sub-Saharan Africa [1]. The disease is caused by parasites of the genus *Plasmodium* and, out of the five species infecting humans, *P. falciparum* causes the most severe form of malaria and accounts for the greatest number of deaths [2]. *P. falciparum* has a complex life cycle involving two different hosts, *Anopheles* mosquitoes and humans. The infection starts with the injection of parasite sporozoites into the human bloodstream, by the bite of an infected mosquito. Sporozoites travel through the blood vessels to the liver to invade hepatocytes and initiate the pre-erythrocytic stage of the disease. Once mature, infected hepatocytes burst releasing exoerythrocytic merozoites that invade erythrocytes, starting the cyclic asexual blood stages. After erythrocyte invasion the parasite develops through three main morphological stages in each 48-hour cycle —the ring, trophozoite and schizont stages— finally replicating into 16–32 daughter merozoites that invade new erythrocytes upon parasite egress [3]. The asexual stages are responsible for malaria clinical symptoms, but some parasites sexually commit and differentiate into male and female gametocytes. Mature sexual forms ingested by mosquitoes undergo several transformations within the Anopheline vector and ultimately infect new human hosts during subsequent blood meals [4].

The malaria parasite relies upon the hexosamine biosynthetic pathway (HBP) for survival during the asexual blood stages [5,6]. Recent research highlights that the HBP is also important during the liver stage development of murine malaria parasite [7]. The end product of the HBP is uridine diphosphate N-acetylglucosamine (UDP-GlcNAc), a crucial sugar nucleotide used by glycosyltransferases to synthesize GlcNAc-containing glycoconjugates [8]. Given the limited scope for protein and lipid glycosylation in *P. falciparum*, UDP-GlcNAc appears to serve exclusively as a precursor for the generation of *N*-glycans and glycosylphosphatidylinositol (GPI) glycolipids (S1 Fig) [9,10]. Some previous work reports that *O*-GlcNAcylation, a dynamic

post-translational modification involved in cellular regulation and signalling, is also present in *P. falciparum*, although the enzymes responsible for this modification have not been identified [11]. *N*-glycosylation is a post-translational modification which involves the synthesis of a glycan donor initiated by GlcNAc-dependent glycosyltransferases. This precursor is then transferred to proteins in the endoplasmic reticulum (ER) by an oligosaccharyltransferase [12]. *P. falciparum* produces highly truncated *N*-glycan donors containing only one or two GlcNAc residues [13]. However, proteins with these minimal *N*-glycans have not been fully characterized [14]. On the other hand, GPI glycolipids are the best characterized and most prominent glycoconjugates in the malaria parasite [15]. GPIs are synthesized through a multistep pathway conserved in eukaryotes, beginning on the cytoplasmic side of the ER with the transfer of GlcNAc, from UDP-GlcNAc, to membrane-bound phosphatidylinositol (PI). The molecule is then de-N-acetylated and moved to the lumenal side of the ER, where subsequent biosynthetic reactions culminate in GPI glycolipid generation. Finally, the protein precursor of GPI-anchored proteins is processed by GPI-transamidase for GPI attachment [16,17].

GPIs are attached to the C-terminus of surface proteins, anchoring them to the external leaflet of the plasma membrane [18,19]. *P. falciparum* expresses more than 30 different GPI-anchored proteins along its life cycle, including asexual [20], sexual [21], sporozoite [22] and pre-erythrocytic stages [23]. Many of these proteins play key roles in host cell adhesion, invasion, and egress. Thus, GPI anchors are considered vital at various stages of the parasite life cycle. Additionally, protein-free GPIs are present on the parasite surface and are four to five times more abundant than GPIs linked to proteins [15,24]. In other protozoan parasites these free glycolipids are involved in motility and cell invasion and are essential for growth [25,26]. Furthermore, free GPIs released into the blood during egress also act as proinflammatory toxins, exerting immunomodulatory effects, which may contribute to the severity of malaria [27,28]. This underpins the importance of these glycoconjugates not only for parasite development through multiple stages but also for malaria pathogenesis.

In this work, we characterized the effects of disrupting the HBP, causing a depletion in UDP-GlcNAc in the asexual intraerythrocytic stages of *P. falciparum*. Our findings show that these disruptions severely impair GPI biosynthesis, leading to the mislocalization of merozoite surface protein 1 (MSP1), a key GPI-anchored protein on the merozoite cell surface. Additionally, the parasite exhibits significant defects in segmentation and completely fails to egress, leading to termination of the asexual life cycle.

## Results

### *Pf*GNA1-depleted parasites fail to progress to the next developmental cycle

*P. falciparum* synthesizes the amino sugar UDP-GlcNAc through a classical HBP involving five enzymatic steps (Fig 1A and S1 Table). This metabolite is key for the production of GPI anchors and *N*-glycans in the malaria parasite [9]. In a previous study, a rapamycin-inducible conditional knockout was engineered in *P. falciparum* targeting glucosamine-6-phosphate N-acetyltransferase (*Pf*GNA1) (S2A Fig), the enzyme responsible for the acetylation of glucosamine 6-phosphate in the HBP [5]. Parasite growth was arrested after three intraerythrocytic developmental cycles (IDC) following rapamycin-induced *gna1* gene excision, revealing that both this enzyme and the HBP are critical for asexual blood stage growth [5]. To determine the specific cause of growth arrest and parasite death following *gna1* disruption, we synchronized parasites to a 5-hour window and treated them with rapamycin immediately after synchronization at the early ring stage (Fig 1B). To assess the dynamics of *gna1* gene excision, we performed PCR amplification of the *gna1* locus after treating tightly synchronized early ring-stage parasites (cycle 0) with 10 nM rapamycin. At 30 hours post-treatment, a significant amount of the non-excised 1546 bp PCR product was still detected, indicating that the gene remained partially intact at least during the trophozoite stage. Complete excision, marked by the exclusive presence of a 738 bp product, was observed from 42 hours post-treatment in cycle 0 (Fig 1C). This excision pattern likely explains the partial parasite growth during the transition from cycle 0 to cycle 1 (S2B and S2C Fig), despite the dramatic reduction of *gna1* transcripts (S3 Fig). In cycle 0, the continued presence of the *gna1* gene —and *Pf*GNA1 enzyme— throughout most of this cycle, combined with the expected existence of a residual pool of UDP-GlcNAc [29], may allow for limited parasite development.

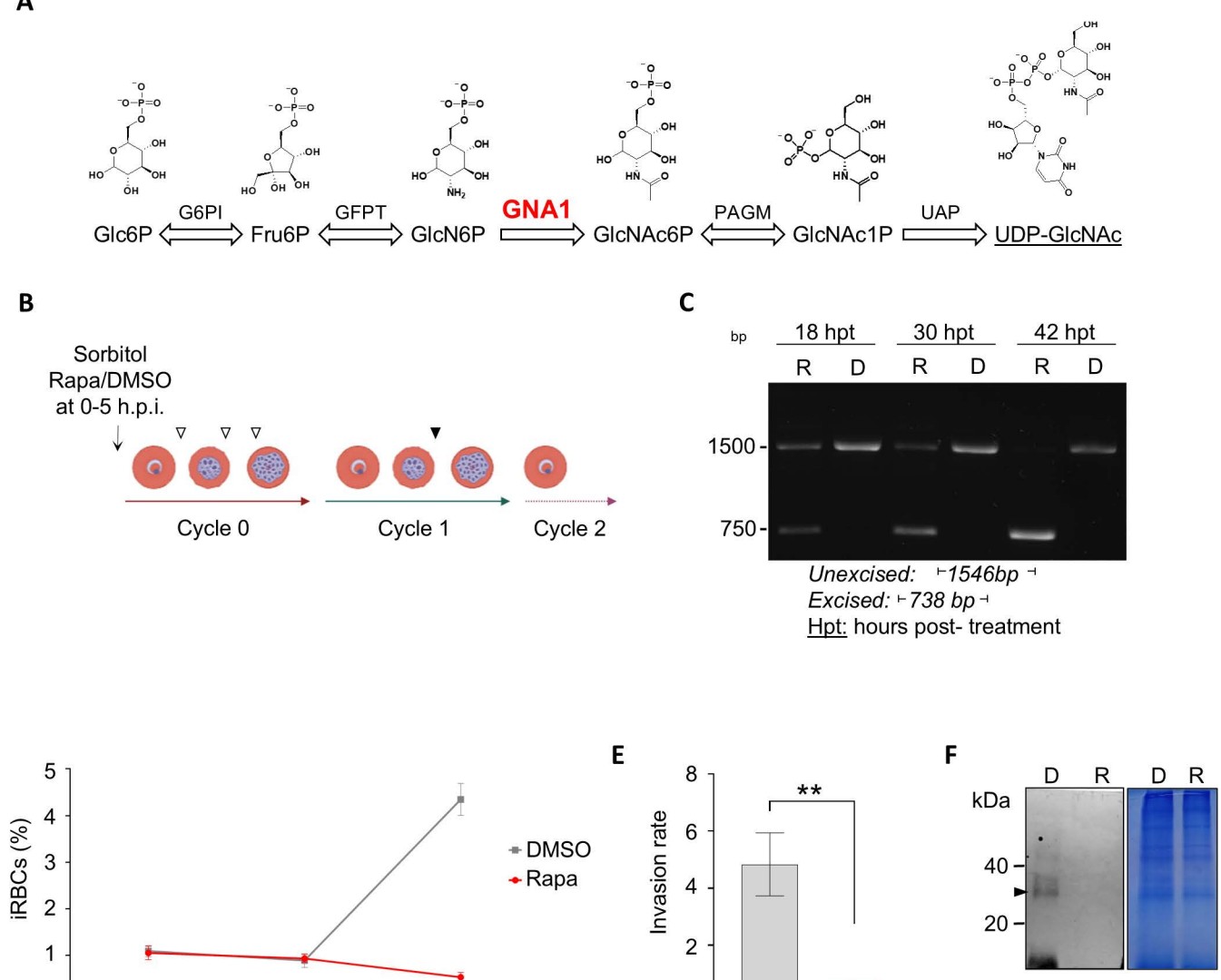

**Fig 1. *Pf*GNA1 is essential for parasite growth during asexual development.** A) Diagram illustrating the enzymatic reactions involved in UDP-GlcNAc biosynthesis through the Hexosamine Biosynthetic Pathway. Glucosamine-phosphate N-acetyltransferase (GNA1) converts glucosamine-6-phosphate (GlcN6P) to N-acetylglucosamine-6-phosphate (GlcN6P). The chemical structures of the pathway intermediates are shown. B) Scheme outlining the timing of rapamycin (or mock, DMSO) treatment, administered at cycle 0 after tight synchronization of parasites within a 5-hour window. The times at which samples were collected for gene excision (white arrowheads) and protein expression (black arrowhead) analyses are also shown. C) PCR analysis of *gna1* was conducted at different times post-DMSO (D) or rapamycin (R) treatment, using primers P5 and P6 specified S2A Fig and S2 Table. D) Parasite growth across cycles 1 and 2 following *Pf*GNA1 disruption assessed by flow cytometry. E) Invasion rates for parasites treated with either DMSO or rapamycin were measured during the transition between developmental cycle 1 and 2. F) Western blot (left) with anti-HA antibody and Coomassie-stained gel (right) as loading control showing protein from trophozoites at cycle 1, treated with DMSO (D) or rapamycin (R) during cycle 0. The arrowhead indicates a band of approximately 30-35 kDa, efficiently depleted upon rapamycin addition. In panels D and E the graph shows mean ± SD values of three independent biological replicates. Statistical analyses were performed using unpaired *t* test. *, P < 0.05; **, P < 0.01; ***, P < 0.001. Abbreviations: G6PI: Glucose-6-phosphate isomerase; GFPT: Glucosamine-fructose-6-phosphate aminotransferase; GNA1: Glucosamine-phosphate N-acetyltransferase; PAGM: Phosphoacetylglucosamine mutase; UAP: UDP-N-acetylglucosamine pyrophosphorylase.

As a result, parasite growth is only marginally affected during cycle 0, with the most pronounced defects, including complete developmental arrest and total loss of viability, occurring in cycle 1. Remarkably, although *gna1* was completely excised by the start of cycle 1, the parasites were still able to mature into late trophozoites and schizonts (Fig 1D). Nevertheless, these parasites were unable to expand from cycle 1 to cycle 2, showing an invasion rate lower than 1 (Fig 1E). Notably, 100 μM GlcNAc supplementation at the beginning of cycle 1 rescued rapamycin-treated parasites, supporting the essential enzymatic role of *Pf*GNA1 in UDP-GlcNAc biosynthesis and survival (S4 Fig).

To study *Pf*GNA1 expression, we engineered the Il3 *gna1*-3xHA-*loxP* line, a new strain bearing a triple hemagglutinin A (HA) epitope at the N-terminus region of the enzyme (S5 Fig) in which the *gna1* locus could still be excised in a regulatable manner by the addition of rapamycin. Consistent with the full disruption of *gna1* by the end of cycle 0, we confirmed the absence of *Pf*GNA1 in cycle 1 by Western blot (Fig 1F).

## Metabolic profiling reveals significant disruptions in the HBP and related intermediates

To confirm that *Pf*GNA1-depleted cells exhibited alterations in the HBP, we conducted metabolic profiling of cycle 1 trophozoites (Fig 2A). This analysis included rapamycin-treated, DMSO-treated (control), and the parental cell lines. Disruption of the HBP at the GNA1 enzymatic step led to a substantial reduction in UDP-GlcNAc levels, the pathway's final product (Fig 2B). Importantly, levels of the intermediates, GlcNAc-1-phosphate and GlcNAc-6-phosphate, downstream of *Pf*GNA1, were also reduced. However, our experimental setup could not distinguish between these two phosphorylated metabolites (Fig 2C). These metabolic changes were evident when comparing both the mock-treated and parental cell lines to the rapamycin-treated line (S6 Fig). Additionally, we observed a significant accumulation of GDP-mannose, a key precursor for the synthesis of dolichol-phosphate-mannose, the mannose donor in GPI anchor biosynthesis (Fig 2D). This accumulation may suggest a reduced availability of glucosamine-phosphatidylinositol (GlcN-PI) as mannose acceptor in the GPI biosynthetic pathway.

Furthermore, parasites treated with either rapamycin or DMSO (control) were grown, and total lipids were extracted and analysed by electrospray-tandem mass spectrometry (ES-MS-MS). Lipidomic analysis revealed an accumulation of phosphatidylinositol (PI) molecules, relative to phosphatidylinositol phosphate (PIP) species. This finding could be explained by the depletion of UDP-GlcNAc, which utilizes PI as an acceptor in the first steps of GPI biosynthesis (S7 Fig). Evidence of the presence of either GlcN-PI or GlcNAc-PI species in the WT was also investigated, however none was observed as it was likely below the level of detection.

## HBP disruption disturbs GPI biosynthesis and alters MSP1 localization in malaria parasites

GPI anchors rely upon UDP-GlcNAc for their biosynthesis and are the most abundant glycoconjugates in *P. falciparum*, thereby playing a crucial role in parasite viability [15,30,31]. Given that UDP-GlcNAc depletion coincided with the accumulation of other upstream GPI biosynthetic precursors, such as GDP-Man, required for mannosylation reactions, we focused our analysis on GPI anchor molecules. Furthermore, the relatively rapid death of parasites following *Pf*GNA1 depletion suggested that *N*-glycans are unlikely to be involved in the observed phenotype. This is because the inhibition of *N*-glycans —another class of GlcNAc-containing glycoconjugate [14]—leads to a delayed death phenotype [32]. Indeed, *Pf*GNA1-disrupted parasites arrest prior to any detectable effect of tunicamycin, a specific inhibitor of *N*-glycosylation [33], suggesting that the loss of *Pf*GNA1 function precedes and is likely independent of *N*-glycosylation related pathways (S8 and S9 Figs). In contrast, GPI quantification revealed a striking reduction in the level of total GPI molecules in cycle 1 rapamycin-treated schizonts (Fig 3A), with a less pronounced, but still detectable decrease already apparent in cycle 0 schizonts (S10 Fig).

The surface of the malaria parasite is densely populated with GPI-anchored proteins, with MSP1 (PF3D7_0930300) being a major component, accounting for approximately one-third of these GPI-anchored proteins on the merozoite

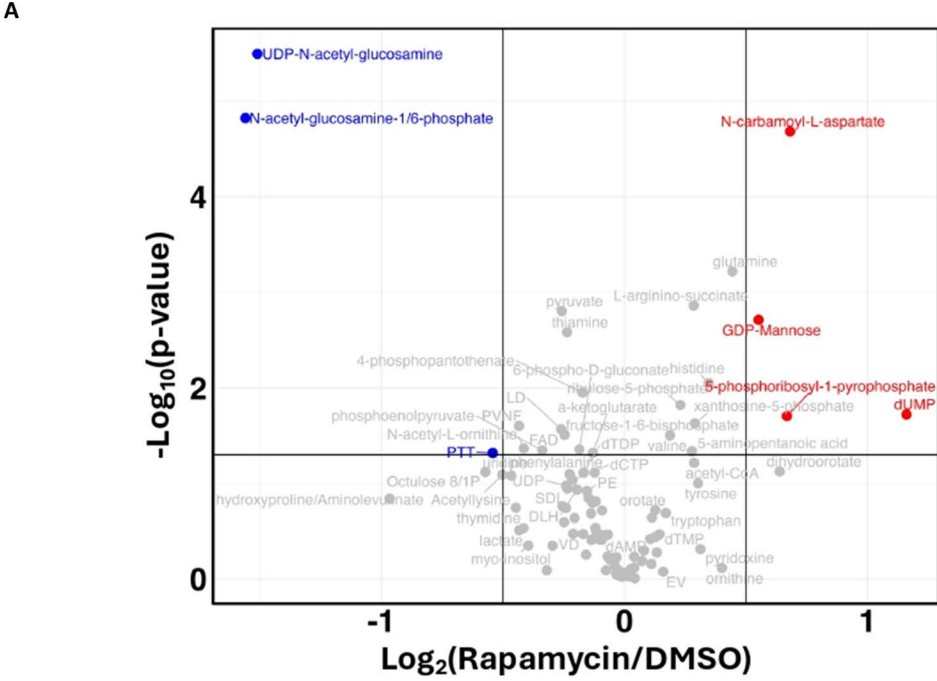

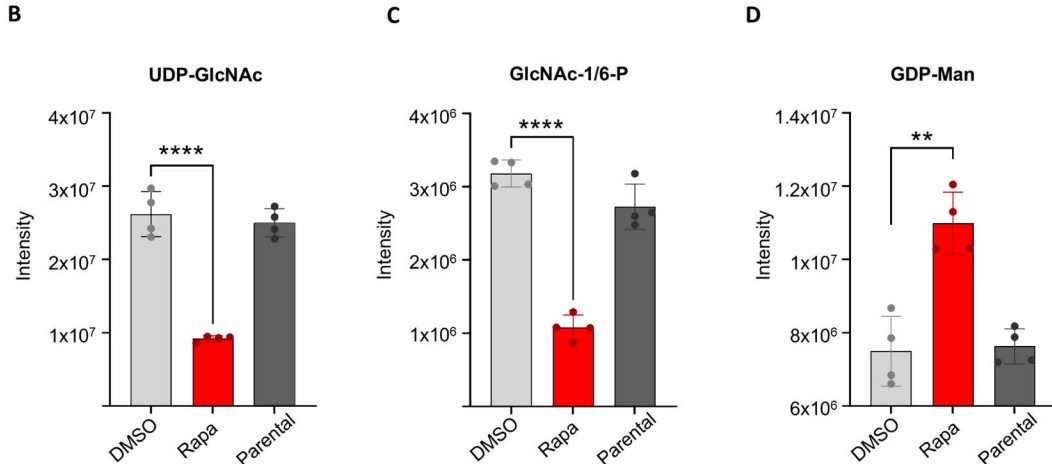

**Fig 2. *Pf*GNA1-disrupted parasites show depletion of HBP intermediates and UDP-GlcNAc.** A) Volcano plot showing global metabolomic changes between *Pf*GNA1-disrupted and DMSO-treated parasites. Detailed analyses of GlcNAc-1/6-P (B), UDP-GlcNAc (C) and GDP-Man (D) levels in *Pf*GNA1-disrupted parasites (treated with rapamycin), non-disrupted parasites (DMSO-treated) and the parental cell line. Data represent the mean and standard deviation from four independent replicates. *p*-values from two-sided Student's *t*-tests are shown in B, C and D, comparing the specified conditions. *, $P < 0.05$; **, $P < 0.01$; ***, $P < 0.001$; ****, $P < 0.0001$. Abbreviations: GlcNAc-1/6-P, *N*-acetylglucosamine-1/6-phosphate; UDP-GlcNAc, uridine diphosphate *N*-acetylglucosamine; GDP-Man, Guanosine diphosphate mannose.

surface [20]. MSP1 is essential for the *P. falciparum* asexual stages, facilitating both invasion and egress [34–36]. Given the reduced GPI anchor content in *Pf*GNA1-disrupted parasites, we investigated MSP1 localization using immunofluorescence microscopy. In DMSO-treated parasites, MSP1 was found to encircle the merozoites in mature schizonts, based upon its location around the Hoechst-stained merozoite nuclei. However, in most *Pf*GNA1-disrupted parasites, MSP1

**A**

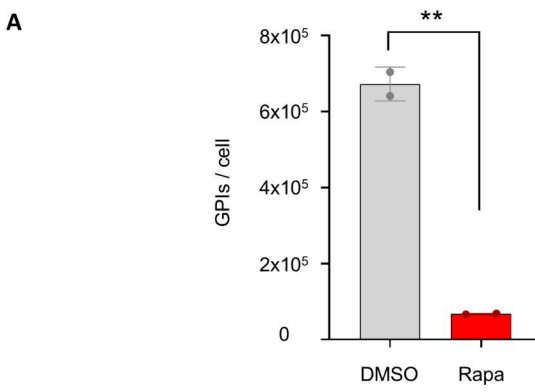

**B**

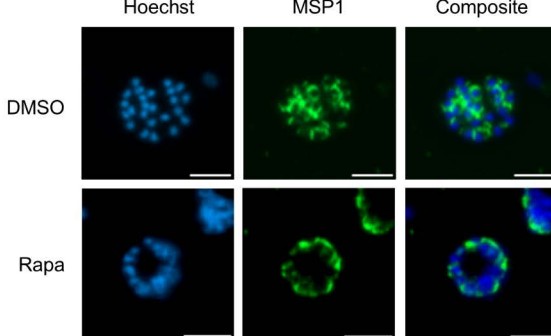

**C**

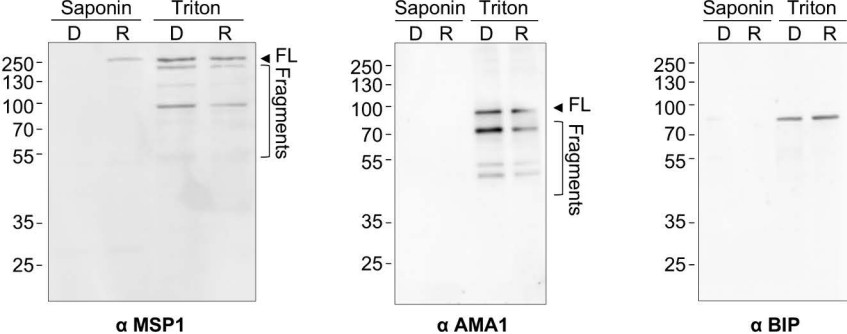

**Fig 3. GPI synthesis is altered in *Pf*GNA1-deficient parasites, disrupting the localization of the GPI-anchored protein MSP1.** A) Quantification of GPI molecules per cell in DMSO- and rapamycin-treated segmented schizonts from cycle 1. B) Immunofluorescence microscopy showing MSP1 distribution in segmented schizonts. MSP1 was labelled with a mouse anti-MSP1 antibody (green), and nuclei were stained with Hoechst 33342 dye (blue). Scale bar is 5 µm. C) Subcellular fractionation of schizonts treated with DMSO or rapamycin was performed sequentially using saponin, followed by Triton X-100 extraction. Both fractions were analysed by SDS-PAGE and Western blotting, using anti-MSP1 (left), anti-AMA1 (middle) and anti-BiP (right) antibodies, respectively. Bands migrating at varying heights, likely reflecting initial MSP1 and AMA1 processing, are observed and indicated in the Triton lanes (left panel). FL: Full-length. Panels B and C display representative images from four independent biological replicates.

appeared to be untethered, shifting away from the merozoite nuclei and diffusing towards the periphery of the schizont plasma membrane (Fig 3B). However, a small subset of *Pf*GNA1-disrupted parasites appear to retain the typical MSP1 distribution observed in untreated parasites (S11 Fig).

Our results indicated that alterations in GPI anchor content, due to *Pf*GNA1 disruption, partially affected the localization of GPI-anchored MSP1. We then hypothesized that non-anchored and mislocalized MSP1 might be secreted into the parasitophorous vacuole (PV) lumen, based upon the immunofluorescence microscopy observations. To investigate this, we treated parasites with saponin, a detergent that selectively permeabilizes the erythrocyte and parasitophorous vacuole membrane (PVM), allowing extraction of soluble proteins present in those compartments without affecting the parasite plasma membrane. The remaining pellets were further extracted with Triton X-100. Western blot analysis revealed that, in *Pf*GNA1-disrupted parasites, MSP1 partially leaked into the PV, suggesting secretion to the peripheral space. In contrast, no such leakage was observed in DMSO-treated controls (Fig 3C, left panel). Additionally, MSP1 processing appeared reduced in *Pf*GNA1-disrupted parasites compared to DMSO-treated parasites [36], as indicated by the weaker processed fragments in the Triton X-100 fraction (S11B and S11C Fig). This observation suggests decreased processing, potentially due to the absence of GPI anchoring. As controls, we also conducted Western blot analyses for apical membrane antigen 1 (AMA1, PF3D7_1133400), a transmembrane protein, and BiP (PF3D7_0917900), a protein member of the endoplasmic reticulum chaperone complex, to confirm that their localization remained unaffected by the disruption of *Pf*GNA1 and GPI biosynthesis. Both proteins were detected in the expected fraction after protein extraction (Fig 3C, middle and right panel, respectively).

### *Pf*GNA1 disruption prevents egress and reinvasion of *P. falciparum*

To determine what impedes parasite growth during cycle 1, we synchronized parasites over a 5-hour window using sequential 70% Percoll centrifugation followed by sorbitol lysis and monitored their development using flow cytometry and light microscopy. Flow cytometry revealed DNA replication, marking the transition from ring (77 hours post-Percoll) to trophozoite and schizont stages (91 hours post-Percoll) [37], in both control and *Pf*GNA1-disrupted parasites (Fig 4A). However, upon careful observation of Giemsa-stained preparations, we found that, although *Pf*GNA1-disrupted parasites appeared to develop into mature stages, they failed to progress further. These parasites often displayed aberrant morphology and did not form fully segmented, multinucleated schizonts with distinct nuclei (Fig 4B). Additionally, these parasites failed to egress, as evidenced by live-cell imaging (S1 Movie), which showed a clear egress defect in rapamycin-treated parasites compared to controls, and by the persistent presence of mature parasites during the transition between cycles 1 and 2, and the lack of new ring forms, as confirmed by flow cytometry (Fig 4C). To further support our observations, we mechanically released the daughter merozoites from mature parasites by passing them through a 1.2 μm filter [35]. Virtually no fully mature merozoites were isolated from *Pf*GNA1-disrupted parasites relative to control parasites (Fig 4D). Moreover, these mechanically released merozoites were unable to reinvade new host red blood cells (S12 Fig). Overall, the disruption of *Pf*GNA1 led to the inability of mature parasites to egress and reinvade new host red blood cells presumably through disruption of the HBP and GPI biosynthesis pathways, thereby halting parasite development.

### HBP and GPI biosynthesis disruption causes segmentation defects and prevents full PVM rupture in *Pf*GNA1-deficient parasites

To further investigate the morphological defects observed in mature schizonts, we performed transmission electron microscopy on rapamycin-induced *Pf*GNA1-disrupted parasites and DMSO-treated controls synchronized at late stages. In DMSO-treated schizonts (Fig 5A), segmentation occurred normally, with each merozoite individually enclosed by a

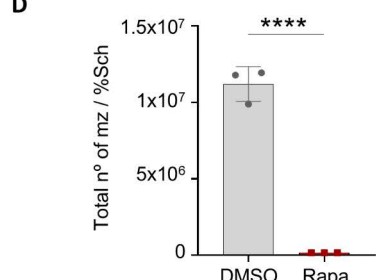

**Fig 4. Disruption of *Pf*GNA1 hints at segmentation defects and prevents parasite egress.** A) Histograms showing a population shift from ring stages to trophozoite/schizont stages. For flow cytometry analysis, the DNA of parasite cultures was stained with SYTO 11, and RBCs were gated using bivariate plots (SSC-H vs. FSC-H), recording 50,000 events within this region. The recorded events were then plotted as a function of fluorescence intensity detected through Filter 1, where subpopulations are distinguished according to their DNA content. B) Microscopy images of Giemsa-stained smears from tightly synchronized (5-hour window) *Pf*GNA1 conditional knockout parasites treated with DMSO (control) or rapamycin. The images show time points during cycle 1 and the transition to cycle 2. Scale bar is 5 μm. C) Egress of synchronized (5-hour window) *Pf*GNA1 conditional knockout parasites treated with DMSO (control) and rapamycin. Total parasitemia and the levels of young forms (rings) and mature forms (trophozoites-schizonts) were measured every two hours by flow cytometry during the transition between cycles 1 and 2, as described in A. The percentage of mature forms at

each time point was normalized to the percentage observed at 91 hpp, when the trophozoite-schizont peak was reached. The ring levels are represented as an absolute number (hpp: hours post-Percoll). D) *Pf*GNA1 conditional knockout parasite, treated with either rapamycin or DMSO, were enriched using Percoll, and merozoites were mechanically released by filtration. The number of merozoites post-filtration was measured by flow cytometry and normalized to the percentage of schizonts in culture before filtration. Panels A and B shows a representative image from four biological replicates. Panels C and D are based on three independent biological replicates, each with technical replicates. The graphs depict the means and standard deviation derived from one representative biological replicate, averaged over technical triplicates. The statistical analysis of panels C and D were performed using unpaired *t* test. \*, $P < 0.05$; \*\*, $P < 0.01$; \*\*\*, $P < 0.001$; \*\*\*\*, $P < 0.0001$.

membrane and exhibiting classical structures, including nuclei and apical organelles. In contrast, *Pf*GNA1-disrupted schizonts displayed clear segmentation defects, with only some schizonts achieving proper segmentation (S13A Fig). Although nuclear division was observed, the developing merozoites appeared conjoined within a single membrane, indicating a failure in the formation of individual membranes required to separate the incipient merozoites (Fig 5B). Some apical organelles appeared well-formed in the mutant strain (Fig 5C), but multiple stacks of membranes were observed in *Pf*GNA1-disrupted parasites (Fig 5D and 5E, arrowheads). Interestingly, in both DMSO- and rapamycin-treated parasites, a uniform contrast across the PVM was observed (Fig 5A and 5C). This suggests that the contents of the red blood cell cytoplasm and the vacuole had mixed, indicating that, although the PVM appears intact, it has likely undergone poration, which is a precursor to PVM rupture, as observed in [38,39].

When parasites were treated with E64, a cysteine protease inhibitor that prevents red blood cell membrane rupture while allowing the breakdown of the PVM [40], it became evident that DMSO-treated schizonts were correctly formed. Single merozoites with well-defined organelles were clearly observed in the absence of the PVM (Fig 5F). A clear change in contrast, resulting from the mixing of vacuolar contents with the host cell cytoplasm, indicated the loss of the PVM as a distinct barrier. Additionally, whorls of membrane vesicles, likely remnants of the PVM [38], were observed, providing further confirmation of this effect (Fig 5F, asterisk). In *Pf*GNA1-disrupted schizonts, partial PVM disruption was observed in some parasites as shown by whorls of membrane vesicles in the RBC cytoplasm (Fig 5G, asterisk). However, in the majority of the observed parasites, schizonts were unable to rupture the PVM (S13B and S13C Fig), suggesting that the egress defect associated with HBP disruption and altered GPI biosynthesis occurs before full PVM rupture and is unrelated to host red blood cell membrane breakdown [36]. Food vacuoles containing internal membranous structures were also present (Fig 5H), with occasional ruptures of the vacuole membrane, releasing hemozoin crystals (Fig 5H, square). These ruptures were further indicated by altered staining patterns with a pH-sensitive fluorescent dye in rapamycin-treated parasites (S14 Fig). The fused merozoites exhibited chromatin condensation at the nuclear periphery, an indicator of irreversible cell damage (Fig 5G and 5J) [41]. Additionally, the ribosomes appeared irregularly distributed throughout the cytoplasm, potentially due to a deficiency in endoplasmic reticulum presence (Fig 5G, 5I, and 5J). Irregular ribosome distribution has also been observed in *Plasmodium* undergoing apoptosis after chalcone treatment. However, this study reported a reduction in ribosome abundance [42], which differs from the increased ribosome presence seen in *Pf*GNA1-disrupted parasites. Thus, despite some features being consistent with programmed cell death changes, further validation is necessary.

In summary, our data shows that *Pf*GNA1 disruption depletes the UDP-GlcNAc pool, which greatly impacts the synthesis of GPI-anchors. Consequently, the localization of MSP1, a key GPI-anchored protein in the membrane of merozoites, is disrupted. Likewise, membrane biogenesis is severely altered during schizont maturation. This results in defects in egress and invasion, thereby halting growth at the schizont stage and ultimately leading to parasite death (Fig 6).

## Discussion

Our data confirms that *Pf*GNA1 disruption leads to a reduction in the HBP intermediates GlcNAc-1-phosphate and GlcNAc-6-phosphate, and to the depletion of UDP-GlcNAc, the final product of the pathway [5]. UDP-GlcNAc acts as donor for GlcNAc-dependent glycosyltransferase reactions. In *P. falciparum* a basic *N*-glycosylation mechanism is

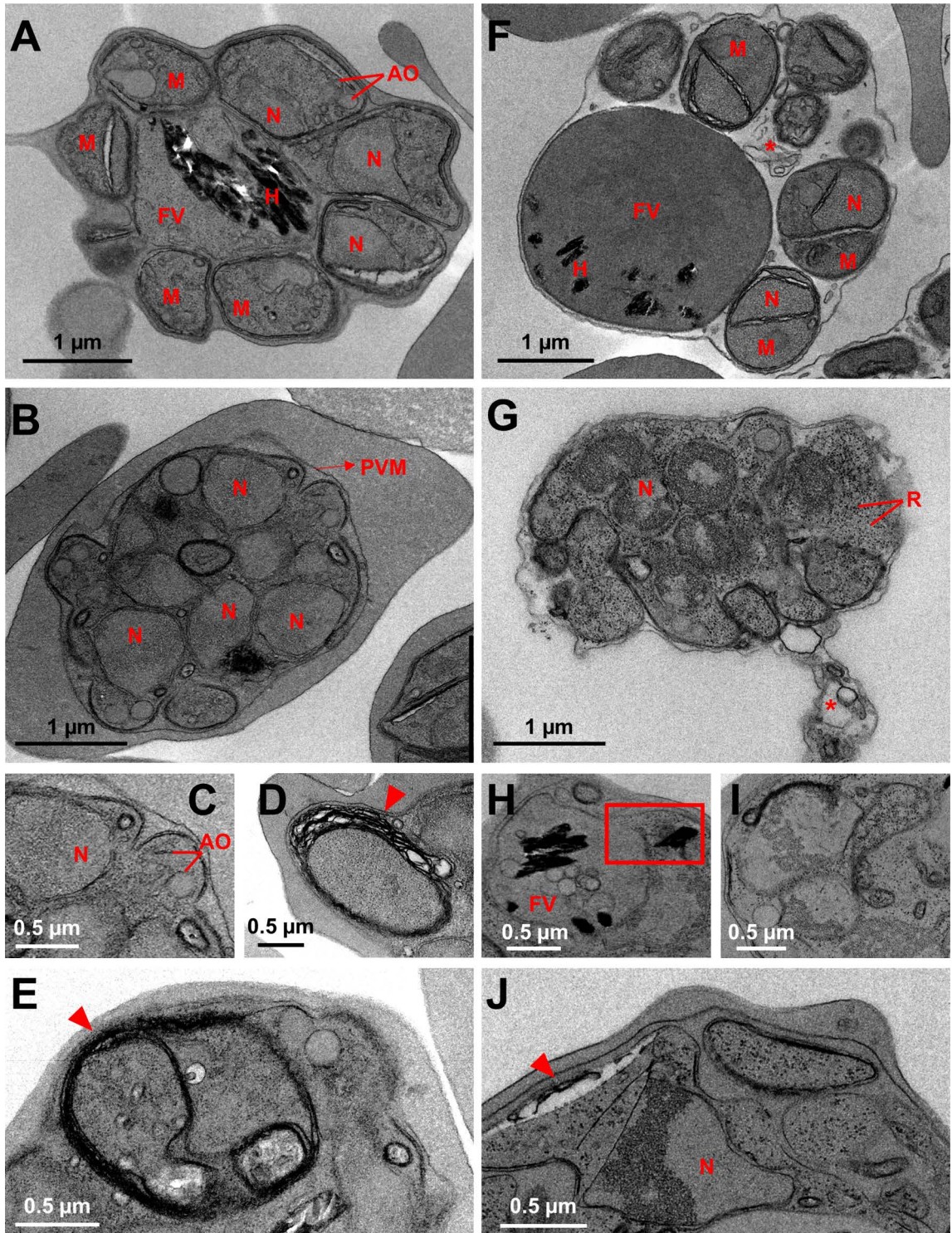

**Fig 5. PfGNA1 disrupted parasites show a severe segmentation defect.** Transmission electron microscopy showing the ultrastructure of PfGNA1 conditional-knockout schizonts treated with DMSO (control) or rapamycin, with or without E64 exposure. A) DMSO-treated parasites show multiple merozoites (M) along with distinct structures, including nuclei (N), apical organelles (AO), and food vacuoles (FV) containing hemozoin crystals (H). B) Rapamycin-treated parasites exhibited multinucleated, fused merozoites enclosed within the parasitophorous membrane (PVM). C) Zoom from B) showing the proper formation of apical organelles. D) and E) Detail of a PfGNA1-disrupted parasite surrounded by multiple layers of stacked membranes

(indicated by arrowheads). F) An E64-treated, DMSO-control parasite, showing isolated merozoites with their respective organelles. Membrane vesicle whorls, marked by an asterisk are also visible. G) An E64-exposed, rapamycin-treated parasite with ribosomes (R) dispersed irregularly throughout the cytoplasm. Multiple nuclei display distinct chromatin condensation. H) Zoom of a FV in a *Pf*GNA1-disrupted parasite as in G), containing internal membranous structures. A square highlights a ruptured FV with liberated hemozoin crystals. I) Detail of a nucleus of an E64-exposed, rapamycin-treated parasite showing chromatin condensation. J) Numerous stacked membrane layers are also observed.

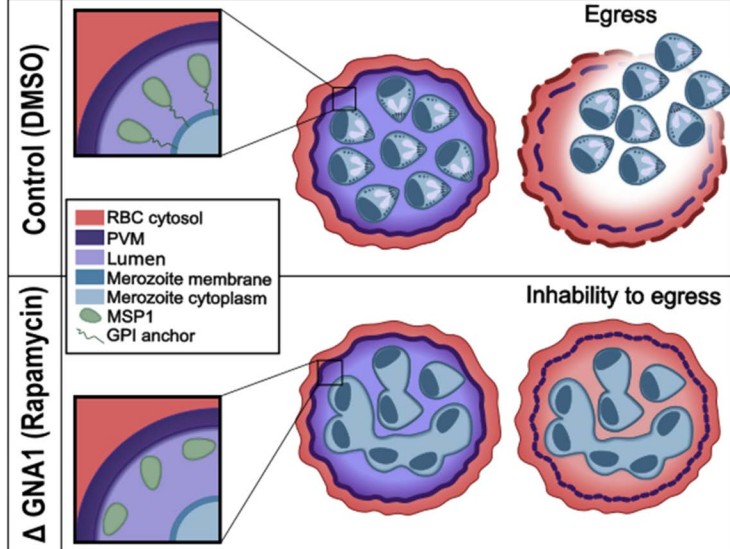

**Fig 6. Diagram depicting impaired parasite egress following disruptions in HBP and GPI biosynthesis.** Rapamycin-treated *Pf*GNA1-disrupted parasites show a marked reduction in GPI molecule levels, compromising MSP1 anchoring to the merozoite membrane and causing it to diffuse away. These parasites also display severe abnormalities during schizont development, characterized by aberrant morphology and the inability to form fully segmented, multinucleated schizonts with distinct nuclei. Furthermore, they fail to fully rupture the PVM, which prevents later breakdown of the red blood cell host membrane and ultimately blocks egress.

conserved although the extent, number, and significance of *N*-glycosylated proteins remain unclear [13,14]. *N*-glycosylation is essential for eukaryotic organisms and, notably, the initial steps of the process involve the action of GlcNAc-dependent glycosyltransferases in the ER [9]. However, disrupting *N*-glycosylation does not immediately halt *P. falciparum* growth. Instead, the parasites continue to grow and invade new red blood cells, where they progress until the early trophozoite stages, at which point their development is halted [32]. This phenomenon, known as delayed death, is commonly, but not exclusively, linked to compounds targeting the apicoplast in *P. falciparum* and related apicomplexans. During delayed death, the disruption of specific cellular functions does not kill the parasite immediately but rather results in the death of its progeny in the following developmental cycle [43–45]. The rapid death of parasites within the first developmental cycle following complete *Pf*GNA1 depletion, coupled with the hallmark stall in schizont stages, strongly suggests that interference with *N*-glycosylation is not responsible for the arrest caused by the alteration of the HBP. Likewise, previous studies have identified a limited number of *O*-GlcNAcylated proteins in the ring and trophozoite stages of *P. falciparum* [11]. Some of these proteins are involved in critical processes, such as glycolysis and chaperone functions, highlighting roles comparable to those of *O*-GlcNAcylated proteins in other eukaryotic organisms [46]. However, similar to other apicomplexan parasites [47], *P. falciparum* lacks known orthologs of the enzymes *O*-GlcNAc transferase (OGT) and O-GlcNAcase (OGA), which are needed for regulating this modification [11,48]. Furthermore, culturing parasites with various concentrations of compounds that interfere with *O*-GlcNAcylation does not impact their viability [11]. Taken together, these observations suggest that *O*-GlcNAcylation is not a key factor in the effects resulting from UDP-GlcNAc depletion in the parasite.

UDP-GlcNAc is crucial for initiating the biosynthesis of GPI glycoconjugates, which are abundant on parasite surfaces, either as free molecules or tethering key proteins to the membrane [15]. Our data shows that UDP-GlcNAc depletion leads to a major reduction in GPI molecules. This depletion is accompanied by a concomitant accumulation of GDP-mannose and phosphatidylinositol species, which are unused precursors in the GPI biosynthetic pathway [17,49]. The reduced availability of GPI anchors results in the mislocalization of MSP1, a major GPI-anchored protein on the surface of *P. falciparum*. MSP1, along with MSP2, constitutes approximately two-thirds of all GPI-anchored proteins on the merozoite surface [20]. Without sufficient GPI anchors, MSP1 is released into the parasitophorous vacuole via the secretory pathway, as it can no longer remain tethered to the parasite membrane [36]. The mislocalization of MSP1 is consistent with findings in other studies, where disruptions in GPI anchor biosynthesis during *P. falciparum* asexual blood stages led to comparable effects [50,51]. Similar alterations have also been observed in the related apicomplexan parasite *Toxoplasma gondii*, where HBP disruption also impairs GPI formation, resulting in the aberrant localization of key GPI-anchored proteins and affecting parasite survival [52]. The dramatic reduction in GPI glycoconjugates in *Pf*GNA1-disrupted parasites, along with the persistence of typical MSP1 distribution in some parasites, suggests that malaria parasites may prioritize preserving crucial GPI-anchored proteins like MSP1, while sacrificing free GPIs and less essential GPI-anchored proteins. This likely helps sustain the intraerythrocytic life cycle, despite disruptions to HBP/GPI biosynthesis ultimately stall asexual development. Moreover, the severe depletion of freshly biosynthesized GPI anchors may also cause proteins that are normally GPI-anchored to remain associated with the ER membrane, either through interactions with membrane-bound proteins or due to incomplete processing by the GPI-transamidase complex. Thus, our findings demonstrate that hindering the HBP severely impacts GPI anchor biosynthesis, and underscore the importance of these glycoconjugates for the proper localization and function of essential *P. falciparum* proteins.

During the schizont stage, the parasite undergoes segmentation, dividing into multiple daughter merozoites [37]. This process involves the parasite plasma membrane invaginating and encapsulating the emerging daughter cells, which eventually detach from the remnants of the mother cell [53]. Disruption of the HBP and GPI anchor biosynthesis leads to major segmentation defects, resulting in abnormal merozoite formation. Interestingly, these segmentation defects closely mirror those observed following the partial rescue of asexual parasites after apicoplast inhibition, which also led to GPI depletion [50]. This partial rescue likely affects the metabolism of cis-polyisoprenols and dolichols, which may potentially alter membrane biophysical properties and/or affect parasite development in later stages. However, given that our data stem from the targeted disruption of the HBP, these findings strongly suggest that the observed segmentation defects are not related to changes in these lipid pools, but rather to the resulting disruption of GPI biosynthesis. Thus, a specific, yet unidentified GPI-anchored protein or group of proteins, potentially including their GPI moieties, may play a significant role in the segmentation mechanism during asexual stages. This idea is reinforced by observations in *P. falciparum* sporozoites, where the GPI anchor of the circumsporozoite protein has been linked to sporozoite budding [54]. Additionally, the analysis of detergent-resistant membranes in asexual stages revealed subsets of GPI-anchored proteins associated with lipid rafts, further suggesting their involvement in critical cellular processes [55]. Finally, the absence of free GPIs on the parasite surface could lead to significant alterations in the plasma membrane, potentially disrupting cytokinesis or other essential processes critical to parasite growth [25].

Disruptions in the HBP and GPI biosynthesis cause malaria parasites to become completely stalled in late schizogony, rendering them incapable of exiting their host red blood cells [50,51]. During egress, newly formed merozoites breach the PVM that encloses them before escaping through the red blood cell membrane [56]. Just prior to PVM rupture, the membrane rounds up [38,39,57] and appears to break at a few initial points before decomposing [38,57], often progressing outward from their breaks [39]. At the same time, a few minutes before rupture, protein kinase G (PKG) is activated by an increase in cyclic GMP levels [58]. This activation triggers the secretion of subtilisin-like protease 1 (SUB1) from parasite exonemes to the parasitophorous vacuole (PV) compartment. SUB1 is thought to play a role in the disassembly of the PVM, although the exact mechanism of membrane rupture remains unclear [56]. SUB1 also facilitates the proteolytic

processing and maturation of serine-repeat antigen (SERA) cysteine proteases, including SERA5 [59], as well as merozoite surface proteins like MSP1 [36]. These effectors bind to and destabilize components of the erythrocyte cytoskeleton, enabling the rupture of the erythrocyte membrane and the subsequent egress of merozoites [36,59]. Our data indicate that GPI-disrupted parasites are stalled just before PVM breakdown, with the electron microscopy data hinting at the presence of small perforations in the PVM [38]. These observations clearly align with recent studies suggesting that a GPI-anchored effector may be crucial for PKG activation and/or PVM rupture [50], while also pointing away from a potential direct role of dolichols in maintaining PVM structure. Overall, alterations in sugar nucleotides or dolichol derivative precursors necessary for GPI biosynthesis highlight the critical roles of these glycoconjugates in parasite development, particularly in mediating PVM rupture and enabling successful merozoite formation, egress and subsequent invasion [50,51].

In summary, our findings demonstrate that the ablation of *Pf*GNA1 disrupts the HBP, significantly impairs GPI production, and halts parasite growth at the mature schizont stage. Given the critical role of GPI-anchored proteins throughout various stages of parasite development [20–23], our study identifies the HBP-GPI anchor biosynthesis metabolic axis as a promising source of potential drug targets against malaria. Furthermore, considering the unique characteristics of *Pf*GNA1, which belongs to a distinct enzyme family with an independent evolutionary origin in apicomplexans [5,6], our work highlights the potential of targeting this enzyme for the development of selective antimalarial compounds.

## Materials and methods

### Ethics statement

Human erythrocytes (anonymized adult blood samples) were obtained from the Banc de Sang i Teixits (Catalonia, Spain), with approval from the Clinical Research Ethics Committee of Hospital Clínic de Barcelona.

### *P. falciparum* culture and transfection

The asexual stages of *P. falciparum* were cultured at 37 °C in an environment consisting of 92% $N_2$, 2% $O_2$, and 5% $CO_2$. Cultures were maintained in RPMI 1640 medium containing Albumax II with washed red blood cells (RBCs) of blood type B+ at a hematocrit of 3–4%. Parasite growth was monitored by counting infected erythrocytes in Giemsa-stained blood smears using light microscopy. The II3 *gna1*-3xHA-*loxP* strain was generated through Cas9-mediated gene replacement. For that, sgRNA and Cas9-expressing construct (pDC2-Cas9-hDHFRyFCU) and linearized pUC19 plasmids were used as backbone. The single guide RNA (sgRNA) targeting the *Pf*GNA1 was chosen using ChopChop [60] and cloned in the pDC2-Cas9-sgRNA plasmid using primers P7/P8. A recodonized version of *gna1* with three hemagglutinin residues in the N-terminus *Pf*GNA1 was cloned in pUC19 (S2A Fig). All primers used are described in S2 Table. *P. falciparum* 3D7 II-3 parasites, which contain the DiCre system inserted into the *p230p* genomic locus (courtesy of Ellen Knuepfer [61]), were transfected during the ring stages. Briefly, after sorbitol synchronization, 200 μl of infected red blood cells (iRBCs) at <5% parasitemia were electroporated with 60 μg of each plasmid at 310V and 950 millifarads. 24 hours later, transfected parasites were selected using 10 nM WR99210 during 5 days, with resistant parasites emerging in culture 28 days after transfection. Clonal parasite lines were subsequently derived from transfected parasite populations through limiting dilution. Recovered parasites were then harvested for genotyping. Genomic DNA was extracted from these parasites using the QIAmp DNA minikit (Qiagen) in accordance with the manufacturer's guidelines. The purified DNA samples were used as templates for PCR amplification of the inserted construct.

### *P. falciparum* growth analysis

To calculate growth curves, tightly synchronized parasites were measured via flow cytometry by incubating 5 μl of parasite culture with 0.75 μl SYTO 11 in 900 μl of PBS for 1 min. Stained parasites were analysed on a BD FACSCalibur where fifty thousand events were recorded for each sample. The invasion rate was calculated as the ratio of new iRBCs (%) to

iRBCs (%) from the previous cycle. To analyse the morphological development of the parasite, images of Giemsa-stained smears were taken on an Olympus BX51 microscope throughout cycle 1 and 2, increasing the frequency of sampling to 4 hours at the end of cycle 1 and beginning of cycle 2.

**Western blot GNA1**

To determine GNA1 expression, the parasites were synchronized to ring-stage by sorbitol and treated for three hours with 10 nM rapamycin or DMSO (used as a vehicle control) to induce the *gna1* excision at cycle 0. Cycle 1 cultures, with >5% parasitemia, at trophozoite stages were collected and centrifuged, resuspended in 2 volumes of 0.2% saponin in PBS and incubated on ice for 10 min. Then, 10 ml of PBS were added to each sample and the mixture was centrifuged at 1800 x $g$ for 8 min at 4°C. The supernatants were removed and the saponin treatment was repeated one more time. Pellets were transferred to 1.5 ml vials, washed with PBS, resuspended in 100 µL of lysis buffer (2% SDS, 60 mM DTT in 40 mM Tris HCl pH = 9.0 containing protease inhibitor cocktail, purchased from Sigma). Then 20 µl of Laemmli 6X (final concentration 1x) were added, incubated for 5 min at 95 °C and 15 µl of each sample were separated by sodium dodecyl sulphate polyacrylamide gel electrophoresis (SDS-PAGE) and transferred onto a nitrocellulose membrane (Bio-Rad, 0.45 µm). For this, electro-transfer was performed at a constant 20 V overnight using the Mini Trans-Blot cell module (Bio-Rad) with Dunn's Transfer Buffer (10 mM $CO_3HNa$, 3 mM $CO_3Na_2$ in 20% methanol, pH 9.9) [62]. The following day, the membrane was blocked for 2 hours at room temperature with 5% (w/v) skimmed milk in TBST (10 mM Tris-HCl pH 8.0, 150 mM NaCl and 0.05% Tween 20). The primary antibody (anti-HA antibody purchased from Santa Cruz Biotechnology, cod. 2C-7392) was diluted 1 : 5000 in 5% skimmed milk/TBST and incubated overnight. After this, the membrane was washed 3 times in TBST for 10 min and incubated for 1 hour with a peroxidase-labelled anti-mouse IgG secondary antibody (Cell Signalling Technology, cod. 7076), diluted 1 : 10,000 in 5% skimmed milk/TBST. Following three washing steps with TBST for 10 min, the membrane was developed with a Clarity Western ECL Substrate (Bio-Rad) and visualized in an ImageQuant LAS 4000 mini biomolecular imager (GE Healthcare).

To determine the expression and subcellular localization of MSP1, AMA1 and BIP, rapamycin and DMSO-treated parasites were collected during cycle 1, after incubation with E64 for 3 hours. Schizonts were enriched with Percoll 70%, washed with 1 ml of PBS/PIC and transferred to 1.5 ml vials. After centrifugation, the pellets were resuspended in 6 volumes of 0.15% saponin in PBS/PIC, incubated 10 min at 4 °C and centrifugated (10,000 $g$, 15 min, 4 °C). The resulting supernatants were transferred to new tubes and stored at -80 °C. The remaining pellets were washed twice with 1ml PBS/ PIC and incubated in 1% Triton X-100 in PBS/PIC for 1 hour at 4 °C under continuous and vigorous shaking, to improve protein extraction. Samples were then sonicated on ice (3 pulse x 10 seconds, 10 seconds off between pulses, amplitude 100%), centrifuged (20,000 $g$, 15 min, 4 °C) and the supernatants containing membrane-associated and organelle-specific proteins were recovered and stored at -80 °C. A BCA Protein Assay (Thermo Fisher Scientific, Waltham, Massachusetts, EUA) was performed, and 10 µg of each extract was resolved by SDS-PAGE and transferred onto nitrocellulose membranes (#10600003, Amersham, 0.45 µm). Electro-transfer was carried out at a constant 30 V overnight using transfer buffer (25 mM Tris base, 192 mM glycine in 20% methanol). Membranes were blocked for 1 hour with 5% BSA in PBS at room temperature. Mouse anti-MSP1 MRA-880A (MR4) was diluted 1:1000 in 1xPBS/0.1% Tween20/5%BSA and incubated overnight at 4°C. In the case of rabbit anti-AMA1 [63] and rat anti-BIP (MR4) were diluted 1:1000 in 1xPBS/0.1% Tween20/5%BSA and incubated 1 hour at room temperature. After this, the membranes were washed 3 times with 0.1% Tween20/PBS1x for 15 min at room temperature. Secondary antibodies, goat anti-mouse HRP (#12349, Sigma Aldrich), goat anti-rabbit HRP (# ab6721, Abcam) and goat anti-rat HRP (#A10549, Invitrogen), were diluted 1:1000 in 1xPBS/0.1% Tween20/5%BSA and incubated for 1 hour at room temperature. Following three washing steps with 0.1% Tween20/PBS1x for 15 min and a final rinse with PBS, the membranes were developed with ECL Western Blotting substrate (#32109, Pierce Thermo Scientific) and visualized in in an ImageQuant LAS 4000 mini biomolecular imager (GE Healthcare).

## Harvest of parasites and metabolite extraction for metabolomic analysis

The II3 *gna1-loxP* line was tightly synchronized over a 5-hour window, using sequential 70% Percoll centrifugation followed by sorbitol lysis, and immediately treated with 10 nM rapamycin or DMSO (control) for one hour. Rapamycin or DMSO were then removed from the culture medium and the parasites were incubated until 57 hours post-Percoll when a new sorbitol synchronization was performed at the beginning of cycle 1. The samples were then incubated for about 81–89 hours post-Percoll and then enriched by magnetic-activated cell sorting (MACS).

For High Performance Liquid Chromatography-Mass Spectrometry (HPLC-MS) analysis, $1x10^8$ parasites per replicate were replated in a 6-well plate and incubated for 2.5 hours to allow recovery. At the time of sample collection, the metabolism was quenched through addition of ice-cold PBS. Parasites were pelleted (500 x $g$, 4 °C, 7 min) and metabolites were extracted from the pellet with 1 ml ice cold 90% methanol, vortexed 30 seconds, and centrifuged for 10 min at maximum speed (16000 x $g$) at 4 °C. Samples were treated identically and swiftly to ensure reproducible results. The methanol supernatants were stored at -80 °C until analysis, when they were transferred to fresh 1.5 ml tubes, dried down completely under nitrogen gas flow, and the metabolite residues stored at -70 °C.

## High Performance Liquid Chromatography-Mass Spectrometry (HPLC-MS) analysis

Once ready for HPLC-MS, samples were resuspended in 100 μl HPLC-grade water with 1 μM chlorpropamide internal standard to account for instrument variation. The samples were then analyzed using a Thermo Scientific Q Exactive Plus Orbitrap MS instrument connected to a Thermo Scientific Dionex Ulitmate 3000 LC setup using a Waters XSelect HSS T3 Column XP (100 x 2.1 mm, 2.5 μM) (Waters, 186006151) at 30˚C. Solvent A was 97% water/3% methanol; 15 mM acetic acid; 10 mM tributylamine; 2.5 μM medronic acid, and solvent B was 100% methanol. The samples ran at 0.200 ml/min with the following gradient: 0-5.0 min: 100% A, 0% B; 5.0-13.0 min: 80% A, 20% B; 13.0-15.0 min: 45% A, 55% B; 15.0-17.5 min: 35% A, 65% B; 17.5-21.0 min: 5% A, 95% B; 21.0-25 min: 100% A, 0% B. Ion detection in negative ion mode was performed using a scan range of 85–1000 m/z with a resolution of 140,000 at m/z 200. The experimental sample run order was randomized with pooled quality control (QC) samples and blanks run regularly throughout the sample queue. Raw data for the metabolomic analysis has been submitted to the NIH Metabolomics Workbench under tracking number ID 5425.

## HPLC-MS data analysis

Raw data files from the instrument were converted to centroided.mzML format using MSConvert of the ProteoWizard Software Package [64]. These files were loaded into El Maven [65] for further data processing, including peak calling, alignment, and peak annotation based on expected m/z and the retention time of previously run metabolite standards. Each El-Maven-called peak is visually examined for acceptable gaussian shape and signal intensity, and the areas of curated peaks are exported to Microsoft Excel for further processing and analysis. All raw metabolite signal intensities were corrected for chlorpropamide variation, and the average blank signal for each metabolite was subtracted from experimental conditions (or substituted for experimental conditions if blank average signal was higher than experimental signal). Metabolites were then filtered for reproducibility in HPLC-MS detection using the relative standard deviation (RSD) of the pooled QC samples, discarding any metabolites with a QC RSD > 30. The resulting feature matrices were then exported and further analyzed using tools on MetaboAnalyst.ca and/or the MetaboAnalystR package for R [66,67].

## Lipid analysis

For ES-MS-MS, parasites were harvested as explained above. $2.6x10^8$ (Parental), $1.9x10^8$ (DMSO) and $1.8x10^8$ (rapamycin) parasites per replicate were replated and incubated for 1 hour to allow recovery. Parasites were collected and resuspended in 2 volumes of 0.2% saponin in PBS, and incubated on ice for 10 min. Then, 10 ml of PBS was added to

each sample and the mixture was centrifuged at 1800 x $g$ for 8 min at 4 °C. Supernatants were removed and the saponin treatment was repeated one more time. Pellets were transferred to 1.5 ml vials and washed with PBS. Total lipids from parasites were subjected to Bligh-Dyer extraction [68]. Briefly, cells were suspended in 100 μl PBS and transferred to a glass tube where 375 μl of 1:2 (v/v) $CHCl_3$: MeOH were added and vortexed. The samples were agitated vigorously for a further 10–15 min. Samples were made biphasic by adding of 125 μl of $CHCl_3$ followed by vortexing. Then, 125 μl of water was added and the samples were vortexed again. After centrifugation at 1000 $g$ at room temperature for 5 min, the lower phase (organic) was transferred to a new glass vial and the upper phase was re-extracted with fresh $CHCl_3$. The resultant lower phase lipid extract was dried under nitrogen and stored at 4 °C until analysis.

Lipid extracts were analysed on a Thermo Exactive Orbitrap mass spectrometer and a ABSciex 4000 QTrap ES-MS-MS. Both positive and negative scans were conducted over various ranges spanning 150–1500 m/z. Lipid identities were confirmed by accurate mass and collision induced fragmentation.

## GPI quantification

For GPI quantification, segmented schizonts at 40 hours post-infection from cycle 0 and cycle 1 were isolated by MACS, as described above. For cycle 0, $3.3 \times 10^8$ (DMSO) and $2.9 \times 10^8$ (Rapamycin) parasites per replicate were replated and incubated for 1 hour to allow recovery. For cycle 1, $4.1 \times 10^8$ (DMSO) and $1.9 \times 10^8$ (Rapamycin) parasites per replicate were similarly replated and incubated under the same conditions. After recovery, cultures were centrifuged at 500 x $g$ for 7 min at 4°C and washed twice with cold PBS. Pellets were stored at -80 °C until analysis.

The method utilises the quantification of all GPIs molecules (protein-free GPIs and GPI-anchored proteins) in a cell by conversion of their GlcN to [1-2H]-2,5-anhydromannitol (AHM) as described elsewhere [69]. Briefly, the internal standard of *scyllo*-inositol (10 ml of 10 mM) was added to each replicate sample of the freeze-dried cells. Various controls with various GlcN and myo-inositol concentrations were prepared and processed in parallel. Samples were subjected to alkaline hydrolysis, by resuspending in 200 μl concentrated aqueous $NH_3$/40% propan-1-ol (1:1, v/v) and incubated at room temperature for >16 h. Samples were subsequently dried under a stream of nitrogen.

Dried samples were dissolved in 30 μl 300 mM sodium acetate buffer. To each sample, 20 μl freshly prepared 1 M sodium nitrite is added and incubated at room temperature for 3 h. Following this nitrous acid deamination, the samples had to be reduced by the sequential addition of 10 μl of 400 mM boric acid followed by ~75 μl 2 M sodium hydroxide and 40 μl freshly prepared 1 M sodium borohydride and left at to reduce at 4°C for 16 h. The samples were desalted using AG50WX12 resin and dried three times with 100 μl 2% acetic acid in methanol to remove the boric acid and once with 200 μl methanol.

Samples were subjected to methanolysis by adding 50 μl of dry methanol containing 0.5 M HCl, transferring to, and flame sealing in capillaries under vacuum and incubating them at 95 °C for 4 h. Upon opening, pyridine (10 μl) with acetic anhydride (10 μl) were added and left for 30 min at RT, to re-N-acetylate any GlcN. Samples were dried from dry methanol (20 μl) twice prior to derivatization with 10 μl of dry pyridine and with 10 μl of N-methyl-N-trimethylsilyl-trifluoroacetamide. After 10 minutes, 2 μl of the mixture was injected onto Agilent GC-MS (MS detector-5973N, GC-6890) with a SE-54column (30 m × 0.25 mm) at 80 °C for 2 min followed by a gradient up to 140 °C at 30 °C/min and a second gradient up to 265 °C at 5 °C/min and held at 265 °C for a further 10 min. Single Ion monitoring of m/z 272 was selected to detect AHM and m/z 318 to detect both *scyllo*- and *myo*-inositol. Peak areas of the standard curves allows the molar relative response factor to be calculated, allowing quantification of AHM and hence the number of GPI containing molecules per cell.

## Indirect immunofluorescence assays

Parasites were synchronized to ring-stage by sorbitol and treated for one hour with 10 nM rapamycin or DMSO (used as a vehicle control) to induce *gna1* excision. The cultures were incubated until next cycle (cycle 1) when a new sorbitol synchronization was performed. Thirty-eight hours after the second synchronization, 10 μM E64 was added to the media

and incubated during 6 hours. Schizonts were enriched by Percoll 70%, resuspended in 1 ml of compete media and washed three times with PBS. In parallel, an eight-well chamber removable (80841, Ibidi GmbH, Germany) was incubated with 80 µl Concanavalin A (5mg/ml in water, Sigma) per well for 20 min at 37 °C. The Concanavalin A was then removed and the slides were left to dry. Then, 150 µl of schizont suspension were seeded in the pre-treated slides and allowed to settle for 10 min at 37 °C. Unbound RBCs were washed away applying 200 µl PBS per well until a faint RBC monolayer remained and preparations were fixed by adding 200 µl cold methanol (-20 °C) and incubated at -20 °C for 7 min. After washing once with PBS, preparations were blocked by adding 150 µL of 2% BSA in PBS, and incubating them for 30 min at room temperature at 400 rpm, orbital agitation. Preparations were then washed 3 times with PBS and 100 µL of mouse anti-MSP1 8A12 ([70], courtesy of Michael Blackman) diluted 1:50 in 2% BSA/PBS was added and incubated overnight at 4 °C with at 400 rpm. The following day, cells were washed 3 times with PBS to remove the excess of primary antibody. The supernatant was removed and donkey anti-Mouse IgG (H + L) Alexa Fluor 488 (#A21202, Thermo Fisher Scientific, Waltham, Massachusetts, EUA), and Hoechst 33342 (Thermo Fisher Scientific, Waltham, Massachusetts, EUA), diluted 1:100 and 1:5000 in 2% BSA/PBS, respectively, was added to the cultures and incubated for 1 hour at room temperature at 400 rpm protected from light. Preparations were then washed 3 times with PBS to remove excess secondary antibody and the slides were mounted in Vectashield (#H-1000, Vector Laboratories). Images were obtained using a LSM980 Airyscan 2 microscope (Zeiss). To analyse food vacuole integrity, cultures of DMSO or Rapamycin-treated segmented schizonts from cycle 1 were washed with PBS, suspended in a solution containing Hoechst 33342 (1:5000) and LysoTracker Red DND-99 (10 µM, Thermo Fischer) and incubated 30 min at 37 °C. After washing with PBS, parasites were resuspended in 200 µl RPMIc and 10 µl were transferred to a µ-slide 8 well high ibiTreat plate (80806, Ibidi GmbH, Germany) containing 200 µl of RPMIc. Visualization was performed using a Revvity Opera Phenix HCS confocal microscopy. All the obtained images were processed with ImageJ (National Institutes of Health, USA).

## Time-lapse microscopy

Parasites were tightly synchronized using a Percoll gradient followed by sorbitol treatment 5 hours later (cycle 0). Immediately after synchronization, 10 nM rapamycin (or DMSO as control) was added for one hour. Cultures were maintained at 37 °C with daily medium changes for 94 hours post-Percoll, corresponding to the end of developmental cycle 1. At this point, 100 nM ML10 was added to reversibly block parasite egress, and cultures were incubated for 7 hours. Parasites were then purified by Percoll gradient, washed to remove ML10 using pre-warmed RPMIc medium, and resuspended in 100 µl RPMIc. Hoechst 33342 (1:5000 dilution) was added to stain nuclei, and 10 µl of this suspension were transferred into a µ-slide 8-well high ibiTreat plate preloaded with 200 µl of RPMIc. Parasites were imaged using time-lapse differential interference contrast (DIC) microscopy on a Revvity Opera Phenix HCS confocal microscope. Imaging began exactly 10 minutes after ML10 removal, and images were captured every 15 seconds. A total of 82 frames were acquired per sample and exported as AVI files at 5 frames per second for analysis.

## Egress assays

The II3 *gna1-loxP* line was tightly synchronized (5 hours) and treated with 10 nM rapamycin or DMSO (control) for one hour. The cultures were incubated up to 56 hours post-Percoll when a new sorbitol synchronization was performed at the beginning of cycle 1. Parasitemia was measured at 77, 88, 91, 93, 95, 97 and 99 hours post-Percoll by flow cytometry on a BD FACSCalibur. The graph represents the percentage of schizonts at each time point relative to the percentage at the peak —observed 91 hours post-Percoll— as well as the absolute number of rings at each timepoint.

## Merozoite purification

Merozoite purification was conducted as previously described [35]. Briefly, the II3 *gna1-loxP* line was sorbitol synchronized and treated with 10 nM rapamycin or DMSO (control) for one hour to induce *gna1* disruption. Cultures were incubated for

56 hours when 200 nM of ML10 was added to the media. After 17 hours a Percoll 70% synchronization was performed and the suspension of segmented schizonts was washed with 10 ml of washing media and centrifuged at 2200 x *g* for 5 min at room temperature. The supernatant was discarded, the pellet resuspended in 10 ml of fresh washing media and passed through a 1.2 μm filter pre-blocked with 1% BSA/PBS for 20 min. The released merozoites were collected in vials previously blocked with 1% BSA/PBS and the filter was washed with 5 ml of washing media, combining the volume with the previous merozoite suspension. Merozoite suspensions were centrifuged at 2200 x *g* for 15 min at room temperature, and the resultant pellets were resuspended in 100 μl of complete RPMI. Merozoite suspensions were measured by flow cytometry on a BD FACSCalibur by adding 20 μl of Countbright absolute counting beads (LifeTechnologies #C36950) and 2 μL of merozoite suspension in 900 μl of PBS, and staining merozoites with 0.5 μl of SYTO 11 green fluorescent nucleic acid stain for 1 min. The number of merozoites was calculated by the number of SYTO 11 positive population relative to the counting beads, fixing the number of beads at 1000 events.

### Flow cytometry-based invasion assay

To study invasion, merozoites were purified following the protocol described before. Cultures of 30 ml containing approx. 4% schizonts treated with DMSO or rapamycin were passed through a pre-blocked 1.2 μm filter. Isolated merozoites were resuspended in 370 μl of complete RMPI and 50 μl of the suspension was placed per triplicate with fresh RBCs in a final volume of 100 μl and 1% hematocrit. The vials were incubated at 37 °C in a shaker for 20 min and transferred to different wells containing 100 μl RPMI, bringing the hematocrit to 0.5%. The 96 well plate was incubated at 37 °C for 24 hours. Parasitemias were measured on a BD FACSCalibur by mixing 900 μl PBS with 20 μl culture and 0.75 μl SYTO 11. The percentage of infected RBCs related to the initial percentage of pre-filtered schizonts was represented in the graph.

### Transmission electron microscopy

The II3 *gna1-loxP* line was synchronized to ring-stage by sorbitol and treated for one hour with 10 nM rapamycin or DMSO (used as a vehicle control) to induce the *gna1* excision. The cultures were incubated until the start of the next cycle, at which point a new sorbitol synchronization was performed. 26 hours after sorbitol synchronization, the cultures were split into two groups. Two cultures were treated with 10 μM E64 for 15 hours, while the other two were left untreated. Segmented schizonts were enriched by Percoll 70%. The resultant pellet of iRBC was resuspended in 1 ml of washing media, centrifuged at 400 x *g* for 5 min and the supernatant discarded. Segmented schizonts were fixed by resuspending the pellet in 500 μl of cold fixation buffer (2% paraformaldehyde, 2.5% glutaraldehyde in 0.1 M phosphate buffer) and incubated at 4 °C during 30 min in a shaker. After centrifugation at 600 x *g* for 5 min, the samples were washed at 4 °C for 10 min in fixation buffer, and washed four times for 10 min with phosphate buffer 0.1 M pH 7.4 at 4 °C. Then, a solution of 1% osmium tetroxide, 0.8% potassium ferrocyanide and 0.1 M PB pH 7.4 was added to the sample and incubated for 1.5 hours at 4 °C in the dark and washed 4 times for 10 min with double-distilled water at 4 °C to eliminate excess of osmium. After dehydrating the sample with increasing concentrations of acetone, infiltration into the Spurr resin was performed followed by polymerization. Ultrathin 60 nm sections of the resin stub were cut using a Leica UC7 ultramicrotome, stained with Aqueous uranyl acetate and Reynolds lead citrate before observation on a J1010 Transmission electron microscope (Jeol) coupled with an Orius CCD camera (Gatan). Sections were imaged at 80kV. TEM was performed on TEM-SEM Electron Microscopy Unit from Scientific and Technological Centers (CCiTUB), Universitat de Barcelona.

### RT-qPCR

Transcriptional analysis was performed as previously described [71]. Briefly, following tight synchronization, trophozoites at 25–30 hours from cycle 0 and cycle 1 were lysed using TRIzol Reagent (Invitrogen), and total RNA was extracted according to the manufacturer's protocol. RNA was subsequently treated with DNase I (Qiagen), purified using the

RNeasy MinElute Cleanup Kit (Qiagen), and reverse-transcribed into cDNA using the AMV Reverse Transcription Kit (Promega). Transcript levels were quantified by real-time quantitative PCR (RT-qPCR) using the standard curve method with Power SYBR Green reagent (Applied Biosystems) in a QuantStudio 7 Real-Time PCR System (Applied Biosystems). Reactions were performed in triplicate. Thermal cycling conditions consisted of an initial denaturation at 95 °C for 10 minutes, followed by 40 cycles of 95 °C for 15 seconds, 57 °C for 30 seconds, and 60 °C for 30 seconds. The protocol concluded with a final series of steps: 95 °C for 15 seconds, 60 °C for 1 minute, and 95 °C for 15 seconds. Transcript levels of gna1 throughout cycle 0 and cycle 1 were normalized to those of the serine-tRNA ligase gene (seryl, Pf3D7_0717700). Primer pairs used for *gna1* (P9 and P10) and seryl (P11 and P12) are listed in S2 Table.

## Statistical analysis

All graphs in this study were generated using GraphPad Prism version 8.0.2. Statistical analyses were conducted using an unpaired Student's *t*-test to compare the means between two independent groups, with significance defined as a *p*-value less than 0.05.

## Supporting information

**S1 Fig. Glycosylation pathways involving UDP-GlcNAc in *P. falciparum*.** The illustration shows the mechanisms of glycosylphosphatidylinositol (GPI) anchor biosynthesis and *N*-linked glycosylation in the endoplasmic reticulum. Additionally, it includes the presence of a potential *O*-GlcNAc cycling, which would require *O*-GlcNAc transferase (OGT) and *O*-GlcNAcase (OGA), although these enzymes have not been identified in the *P. falciparum* genome. Red dashed arrows indicate enzymatic reactions that depend on UDP-GlcNAc, the end product of the Hexosamine Biosynthetic Pathway (HBP). The diagram also includes GDP-mannose, a sugar nucleotide critical for GPI biosynthesis. (TIF)

**S2 Fig. Cycle 0 to cycle 1 transition in II3 *gna1-loxP* parasites.** A) Diagram of rapamycin-induced site-specific excision. The transgenic strain II3 *gna1-loxP* was generated by CRISPR-Cas9 in a DiCre-expressing strain [5]. The native *gna1* gene was replaced by a recodonized version gen floxed by two *loxP* sites. The addition of rapamycin induces Cre recombinase dimerization, which recognizes the *loxP* sites and removes the sequence between them, inducing *gna1* gene excision. Excision reduces the amplicon from 1,546 bp to 738 bp. The homology regions (HR) used for CRISPR-Cas9-based *gna1* locus engineering are indicated. The hybridization sites of the primers P5 and P6 used to confirm *gna1* excision are also shown. All primers used are described in S2 Table. B) Parasite growth during cycles 0 and 1 following *gna1* gene disruption. The II3 *gna1-loxP* strain was tightly synchronized (5 hours) and treated with rapamycin or DMSO (control) for one hour. Parasitemia was measured immediately (rings, cycle 0) and 60 hours after sorbitol synchronization (rings, cycle 1) by flow cytometry. C) Invasion rates for II3 *gna1-loxP* parasites treated with either DMSO or rapamycin were calculated for the transition between developmental cycles 0 and 1. In panels B and C the graph shows mean±SD values of three independent biological replicates. Statistical analyses were performed using unpaired *t* test. \*, P<0.05; \*\*, P<0.01; \*\*\*, P<0.001. (TIF)

**S3 Fig. Relative transcript abundance of *gna1* normalized to serine-tRNA ligase gene (seryl, reference gene) in DMSO and Rapamycin-treated parasites.** RNAs were extracted from 25-30 hours trophozoites in both, cycle 0 and cycle 1. Transcript abundance was quantified from qPCR data using standard curves and is shown as mean±standard deviation from three technical replicates. A reduction in transcript abundance was observed in the rapamycin-treated parasites, which is more pronounced in cycle 1 than in cycle 0. Values represent one biological replicate per condition. Statistical analyses were performed using an unpaired *t* test. \*\*\*, P<0.001; \*\*\*\*, P<0.0001. (TIF)

**S4 Fig. GlcNAc supplementation during cycle 1 following complete *Pf*GNA1 knockout restores parasite growth, indicating that *Pf*GNA1's essential role in asexual blood stages is enzymatic.** A) Schematic representation of the timing of rapamycin (or mock, DMSO) treatment and GlcNAc supplementation. Rapamycin (or DMSO) were administered for one hour during cycle 0 after tight synchronization of parasites within a 5-hour window. 100 μM GlcNAc was added at the beginning of cycle 1 and maintained until the end of the experiment. The time points at which samples were collected for flow cytometry analysis are also indicated (white arrowheads). B) Parasite growth across cycles 1 and 2 following *Pf*GNA1 disruption, assessed by flow cytometry. C) Invasion rates for parasites treated with either DMSO or rapamycin were measured during the transition from developmental cycle 1–2. In panel C, the graph shows the mean ± SD values of three technical replicates from one representative biological replicate out of three. Statistical analyses were performed using an unpaired *t* test. ****, $P < 0.0001$.
(TIF)

**S5 Fig. Generation and characterization of the asexual cycle of II3 *gna1*-3xHA-*loxP* line A) Diagram depicting the engineering of *P. falciparum gna1*.** Using Cas9-mediated genome editing, the native *gna1* open reading frame was replaced with a recodonized version of the *P. falciparum gna1* gene containing three hemagglutinin (HA) epitopes at the N-terminus, with all the insert flanked by two *loxP* sites. Sequence insertion via double crossover homologous recombination was enabled by Cas9 endonuclease-induced double-strand breaks in the DNA, guided by a single guide RNA (sgRNA) targeting a 20-nucleotide region within *gna1*. Homology-directed repair was enabled by the addition of homologous sequences flaking the *loxP* sites. The HA-tag start codon is marked with a red tick mark with an arrowhead. The positions of the start and stop codons for *gna1* are indicated by a black tick mark with an arrowhead and a red tick mark with a T-cap, respectively. B) PCR detection of transgenic parasites. PCR-based detection of transgenic parasites was performed using P1 and P2 primers, which amplify a 581 bp of the native *gna1*. The parental strain (II3) was included as a control. The primers P3 and P4 were used to amplify a 587 bp fragment corresponding to the recodonized version of *gna1* in transgenic parasites (Clones A8, B6 and F4). The donor plasmid with the recodonized gna1 sequence, used for the generation of the transgenic lines, was included in the PCR as positive control (C+). All primers used are described in S2 Table. C) Parasite growth during cycles 0 and 1 following *gna1* disruption. The II3 *gna1*-3xHA-*loxP* strain was tightly synchronized at a 5-hour window and treated with either rapamycin or DMSO (control) for one hour. Parasitemia was assessed immediately (rings, cycle 0) and 55 hours post-sorbitol synchronization (rings, cycle 1) using flow cytometry. D) II3 *gna1*-3xHA-*loxP* invasion rates were calculated for the transition between developmental cycle 0 and 1 following treatment with either DMSO or rapamycin. E) Parasite growth during cycle 1 and 2 following *gna1* disruption. II3 *gna1*-3xHA-*loxP* rings from cycle 1 were adjusted to 1% parasitemia, and parasite growth throughout cycles 1 and 2 was monitored by flow cytometry. Parasitemia was measured immediately after adjustment (rings, cycle 1), 107 hours post-sorbitol (Trophozoites and Schizonts cycle 1) and 130 hours post-Sorbitol (Rings cycle 2 or Trophozoites and Schizonts cycle 1). F) Invasion rates for II3 *gna1*-3xHA-*loxP* parasites treated with either DMSO or rapamycin were calculated for the transition between developmental cycles 1 and 2. In panels C to F, analyses were performed on three different clones to account for biological variability.
(TIF)

**S6 Fig. Volcano plot showing global, targeted metabolomic changes between *Pf*GNA1-disrupted and parental, non-treated parasites.** A similar trend is observed when *Pf*GNA1-disrupted parasites are compared with DMSO-treated parasites or parental non-treated parasites.
(TIF)

**S7 Fig. Lipid analysis of DMSO or rapamycin treated *Pf*GNA1-disrupted parasites.** Bligh-Dyer extracted lipids were analysed by ES-MS-MS, negative ion mode 820–1100 m/z, showing primarily the PI and PIP species. A) DMSO and B) Rapamycin *Pf*GNA1-disrupted. Lipid identities confirmed by accurate mass and collision induced fragmentation.
(TIF)

**S8 Fig. GlcNAc supplementation during cycle 0 rescues the growth of *Pf*GNA1-disrupted parasites, but the parasites die at the end of the same cycle following GlcNAc withdrawal.** A) Schematic representation of the timing of rapamycin (or mock, DMSO) treatment and GlcNAc supplementation, administered during cycle 0 after tight synchronization of parasites within a 5-hour window. Rapamycin (or DMSO) were added for one hour while 100 μM GlcNAc was added at the beginning of cycle 0 and removed at the end of the same cycle, during the segmented schizont stage. The time points at which samples were collected for flow cytometry analysis are also indicated (white arrowheads). B) Parasite growth across cycles 0 and 1 following *Pf*GNA1 disruption, assessed by flow cytometry. C) Invasion rates for parasites treated with either DMSO or rapamycin were measured during the transition from developmental cycle 0–1. D) Parasite growth across cycles 1 and 2 following *Pf*GNA1 disruption and GlcNAc removal, assessed by flow cytometry. E) Invasion rates for parasites treated with either DMSO or rapamycin were measured during the transition from developmental cycle 1–2. In panels C and E, the graphs show the mean ± SD values of three technical replicates from one representative biological replicate out of three. Statistical analyses were performed using an unpaired *t* test. ****, $P < 0.0001$.
(TIF)

**S9 Fig. Parasite death in cycle 1 schizonts results from HBP disruption, leading to UDP-GlcNAc depletion and GPI reduction rather than *N*-glycosylation defects.** A) Schematic representation of the timing of rapamycin (or mock, DMSO) treatment, GlcNAc supplementation, and tunicamycin treatment. Rapamycin (or DMSO) and GlcNAc were administered during cycle 0 following tight synchronization of parasites within a 5-hour window. Rapamycin and DMSO were added for one hour, while 100 μM GlcNAc was added at the beginning of cycle 0 and removed at the end of the same cycle, during the segmented schizont stage. To dissect the effect of *N*-glycosylation inhibition on DMSO- (mock) or Rapamycin-treated parasites (*Pf*GNA1- and HBP-disrupted), 20 μM tunicamycin was added at the beginning of cycle 1 and maintained until the end of the experiment. Time points for sample collection by flow cytometry (white arrowheads) and Giemsa staining (black arrowheads) are also indicated. B) Parasite growth from cycle 0 to cycle 3 following *Pf*GNA1 disruption, assessed by flow cytometry. C) Giemsa-stained smear images showing that rapamycin-treated parasites (with or without tunicamycin) failed to progress beyond cycle 1 schizonts, while DMSO-treated parasites exposed to tunicamycin died at the trophozoite stage of cycle 2. Thus, after *Pf*GNA1-disruption parasite death is driven by UDP-GlcNAc depletion which halts GPI biosynthesis, rather than by the effect of tunicamycin. Scale bar represents 5 μm.
(TIF)

**S10 Fig. Quantification of GPI molecules per cell in DMSO- and rapamycin-treated segmented schizonts from cycle 0.** A total of $3.3 \times 10^8$ and $2.9 \times 10^8$ parasites per replicate were analyzed for the DMSO and Rapamycin conditions, respectively. GPI molecules per cell were also quantified in $3.3 \times 10^8$ uninfected RBCs per replicate, and the value obtained was subtracted from the parasite culture values to account for GPI molecules derived from the RBCs.
(TIF)

**S11 Fig. Supplementary immunofluorescence and Western blot analyses.** A) Immunofluorescence microscopy showing the typical distribution of MSP1 in less than 6% of *Pf*GNA1-disrupted schizonts. A total of 108 parasites were analyzed. MSP1 was labelled with a mouse anti-MSP1 (green), and nuclei were labelled with Hoechst 33342 (blue). Scale bar represents 5 μm. B) Western blot analysis of MSP1 and quantification of band intensities expressed relative to the control condition, which was set to 100%. Fragments showed a marked reduction in intensity, ranging from approximately 27% to 57% relative to the control. C) Protein bands visualized on the Coomassie-stained gel used as a loading control for the Western blot shown in Fig 3C. A total of 10 μg of protein was loaded for each condition.
(TIF)

**S12 Fig. Invasion assay of merozoites released mechanically.** II3 *gna1-loxP* parasites were treated with rapamycin or DMSO during cycle 0 and incubated until the end of cycle 1. Merozoites were mechanically released from segmented

schizonts and subsequently incubated with RBCs to allow invasion. Parasitemia was measured 24 hours later by flow cytometry and reported as the percentage of infected red blood cells relative to the percentage of schizonts before filtration. Statistical analysis was performed using unpaired $t$ test. *, P<0.05; **, P<0.01; ***, P<0.001; ****, P<0.0001. (TIF)

**S13 Fig.  Transmission electron microscopy of *Pf*GNA1-disrupted parasites.** A) Transmission electron microscopy showing normal segmentation in less than 2% of *Pf*GNA1-disrupted schizonts. A total of 90 parasites were analyzed. From left to right, the images were obtained at 30000x, 50000x and 80000x magnification, respectively. B) Transmission electron microscopy images depicting differences in PVM rupture between *Pf*GNA1 conditional knockout schizonts treated with DMSO (control) and rapamycin, following treatment with E64. Arrowheads indicate parasites with an intact PVM, while asterisks mark those with fully disrupted PVM, evidenced by a lighter, less electron-dense appearance of the cytoplasm after PVM breakdown. Magnification: 5000x C) Comparison of the percentage of fully disrupted PVM in *Pf*GNA1 conditional knockout parasites treated with DMSO (control) or rapamycin. Panel B is representative of more than 200 schizonts. For panel C, approximately 15 images containing over 200 schizonts, were analyzed for each condition. Schizonts were categorized based on whether their PVM was intact or disrupted. The statistical analysis of the boxplot in panel C was performed using unpaired $t$ test. ***, P<0.001. (TIF)

**S14 Fig.  Food vacuole integrity analysis by LysoTracker Red DND-99 staining of DMSO or Rapamycin-treated schizonts from cycle 1.** A) Representative immunofluorescence microscopy images showing Lysotracker staining (orange) in segmented schizonts from cycle 1. Nuclei were counterstained with Hoechst 33342 (blue). LysoTracker Red DND-99 is a fluorescent probe that accumulates in acidic compartments. In DMSO-treated parasites, the food vacuole appeared as a distinct, round-shaped organelle with intense fluorescence signal. In contrast, in rapamycin-treated parasites, the fluorescence was dispersed irregularly throughout the cytoplasm. This pattern suggests disruption of the vacuolar membrane and loss of acidic compartmentalization, likely due to the breakdown of the food vacuole structure. Scale bar is 5 µm. B) Quantification of Lysotracker intensity shown as a scatter plot of individual data points. In rapamycin-treated parasites a reduction in LysoTracker signal intensity is observed. The mean is indicated by a blue line, and the standard error of the mean (SEM) by a red line. At least 500 schizonts were analyzed in each condition. Statistical significance was determined using Welch's t-test. **** P<0.0001. (TIF)

**S1 Movie.  *Pf*GNA1 disrupted parasites display an egress defect.** Synchronized parasites were treated with 10 nM rapamycin (or DMSO) in cycle 0 and allowed to develop until the end of cycle 1. Egress was reversibly blocked during 7 hours using ML10, followed by Percoll enrichment, ML10 washout, nuclei staining with Hoechst 33342 (1:5000) and live imaging. Parasites were observed by time-lapse DIC microscopy starting 10 minutes after ML10 removal, with images captured every 15 seconds. A total of 82 frames were acquired and exported at 5 frames per second. Rapamycin-treated parasites showed a clear egress defect compared to controls. Time in minutes is indicated in the top right corner of each frame. (MP4)

**S1 Table.  Enzymes involved in the Hexosamine biosynthetic pathway (HBP) in *P. falciparum*.** (PDF)

**S2 Table.  List of primers used in this study.** (PDF)

## Acknowledgments

We are very grateful to Michael Blackman for sharing the anti-MSP1 antibody, Viola Introini and Matt Govendir, from Maria Bernabeu's research group for their assistance with confocal imaging, Alfred Cortés and their team for technical advice and reagents and the TEM-SEM Electron Microscopy Unit from Scientific and Technological Centers (CCiTUB), Universitat de Barcelona, and their staff for their support and advice on TEM technique. The authors would like to acknowledge the Huck Institutes' Metabolomics Core Facility (RRID:SCR_023864) for maintenance of the Thermo Exactive Plus.

## Author contributions

**Conceptualization:** Luis Izquierdo.

**Formal analysis:** María Pía Alberione, Gabriel W Rangel, Terry K Smith, Luis Izquierdo.

**Funding acquisition:** Luis Izquierdo.

**Investigation:** María Pía Alberione, Yunuen Avalos-Padilla, Gabriel W Rangel, Miriam Ramírez, Tais Romero-Uruñuela, Àngel Fenollar, Jonathan Ortega-Barrionuevo, Marcell Crispim, Terry K Smith, Manuel Llinás, Luis Izquierdo.

**Methodology:** María Pía Alberione, Gabriel W Rangel, Miriam Ramírez, Tais Romero-Uruñuela, Àngel Fenollar, Jonathan Ortega-Barrionuevo, Marcell Crispim, Terry K Smith, Manuel Llinás, Luis Izquierdo.

**Resources:** Terry K Smith, Manuel Llinás, Luis Izquierdo.

**Supervision:** Luis Izquierdo.

**Writing – original draft:** María Pía Alberione, Luis Izquierdo.

**Writing – review & editing:** María Pía Alberione, Yunuen Avalos-Padilla, Gabriel W Rangel, Miriam Ramírez, Tais Romero-Uruñuela, Àngel Fenollar, Jonathan Ortega-Barrionuevo, Marcell Crispim, Terry K Smith, Manuel Llinás.

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
