## [Decision Letter · Decision Letter 0]

30 Jan 2025

Hexosamine Biosynthesis Disruption Impairs GPI Production and Arrests Plasmodium falciparum Growth at Schizont Stages

Dear Dr. <!--StartFragmentDr Izquierdo<!--EndFragment

Thank you for submitting your manuscript to PLOS Pathogens. After careful consideration, we feel that it has merit but does not fully meet PLOS Pathogens's publication criteria as it currently stands. Therefore, we invite you to submit a revised version of the manuscript that addresses the points raised during the review process.

Please submit your revised manuscript within 60 days Mar 31 2025 11:59PM. If you will need more time than this to complete your revisions, please reply to this message or contact the journal office at plospathogens@plos.org. Please include the following items when submitting your revised manuscript:

We look forward to receiving your revised manuscript.

Kind regards,

Laura Kirkman, MD

Academic Editor

PLOS Pathogens

Tracey Lamb

Section Editor

Editor-in-Chief

PLOS Pathogens

orcid.org/0000-0003-2946-9497

Editor-in-Chief

PLOS Pathogens

orcid.org/0000-0002-7699-2064

**Additional Editor Comments:**

We had the optimal circumstance of multiple reviewers being interested in your paper and as a result were able to obtain five reviews, with differing opinions on what to do moving forward, but also reflecting the general interest in your work. All commented on the quality of the work but several noted that there was concern for the novelty of work given this knockdown line was previously published. Reviewer 5 has suggested some experimental work that would better support your conclusion in this paper that loss of GPI biosynthesis is responsible for the observed phenotype and provide more mechanistic insight.

**Journal Requirements:**

1) We noticed that you used the phrase 'data not shown' in the manuscript. We do not allow these references, as the PLOS data access policy requires that all data be either published with the manuscript or made available in a publicly accessible database. Please amend the supplementary material to include the referenced data or remove the references.

4) Please ensure that the funders and grant numbers match between the Financial Disclosure field and the Funding Information tab in your submission form. Note that the funders must be provided in the same order in both places as well. State what role the funders took in the study. If the funders had no role in your study, please state: "The funders had no role in study design, data collection and analysis, decision to publish, or preparation of the manuscript.".

**Reviewers' Comments:**

Reviewer's Responses to Questions

**Part I - Summary**

Reviewer #1: The main finding “the disruption of PfGNA1 led to the inability of mature parasites to

egress and reinvade new host red blood cells presumably through disruption of the HBP

and GPI biosynthesis pathways, thereby halting parasite development.”

is novel and relevant

Reviewer #2: Strengths/weaknesses of the study, novelty/significance, general conduct and scientific rigour.

The present study is concerned with the inhibition of the biosynthesis of glycosylinositol, a glycolipid that occurs in free and protein-bound forms in the malaria parasite. The protein-bound form serves as an anchor for proteins in the outer leaflet of a membrane. The free forms play a role in the pathogenicity of malaria. In this respect, the present study could contribute to the development of chemotherapy beyond the mere scientific information.

UDP-GlNAc is required as a component of GPIs for biosynthesis. GlcNac is provided via the hexosamine biosynthesis pathway, which starts from D-glucose. By switching off the enzyme glucosamine-6-phosphate N-acetyltransferase which acetylates GlcN-6-P to GlcNAc-6-P, the biosynthesis of UDP-GlcNac is inhibited and, as a consequence, the synthesis of GPIs.

The investigations are carried out using state-of-the-art methods; this concerns the provision of the mutant, the mass spectroscopic investigations of the lipids and the imaging techniques. The new results are the outcome of collaboration between several highly specialized working groups.

The results merit publication.

Reviewer #3: In this manuscript, Alberione et al. have characterized the hexosamine biosynthesis pathway and discovered that PfGNA1 is essential for GPI production, parasite egress and viability. Overall, the study was well designed and supported by good data. There are several issues that should be addressed.

Reviewer #4: It is well established that Plasmodium has limited carbohydrate modifications to its proteins, and an apparent majority of the UDP-GlcNAc pool is used in the synthesis of GPI, which tethers a number of surface proteins to the parasite plasma membrane. In the present study, the authors phenotypically assess the inducible knockout of PfGNA1, which they have previously shown to be essential for parasite replication, presumably through its role in the hexosamine biosynthesis pathway. While the experiments performed here appear to be of high quality, new mechanistic insight is limited.

The authors have already published that inducible deletion of GNA1 blocks parasite replication through the expected biochemical mechanism. They have nice microscopy that shows mislocalization of one GPI-anchored protein (MSP1, which is known to be essential). They also have nice EM showing parasites are nonviable by the time of egress. But given the long time period in assessing phenotype (a full lytic cycle), and the fact that GPI anchored proteins like MSP1 are known to be essential, there isn’t much mechanism to be had here that wasn’t already known. The authors note that a large portion of the GPI in membranes is not anchored to any protein, and suggest it may be required for egress/motility. But no effort was made in the present study to differentiate whether the phenotype is due to a lack of GPI-anchored protein or just free GPI (probably both; I understand that this would be quite difficult to ascertain).

Reviewer #5: This study investigates the consequences of disrupting hexosamine biosynthesis (specifically GlcNAc6P synthesis) in the malaria parasite, Plasmodium falciparum. Rapamycin inducible knock-down of the gene encoding GlcN6P acetyltransferase (GlcN6P-AT) in asexual blood stages leads to parasite death over 1-2 cycles of infection, and correlates with a reduction in intracellular levels of down-stream intermediates in the hexosamine biosynthesis pathway; GlcNAc1/6P and UDP-GlcNAc. Further, disruption of the pathway leads to significant ultrastructural changes in blood stages, defects in the processing of the major GPI-anchored protein, MSP1, and defects in parasite egress. The authors propose that the observed phenotype primarily reflects the loss of GPI biosynthesis and expression/processing of GPI-anchored proteins, rather than loss of other GlcNAc-dependent pathways such as protein N-glycosylation. There is clear interest in understanding the function of different glycosylation pathways in these parasites and the work described in the manuscript is generally well performed. However, the novelty of this study is unclear. The authors have recently published a paper in which the same enzyme is inducibly knocked-out in the same parasite (Ref 5,6), and several other studies have shown that enzymes/pathways involved in GPI biosynthesis are essential in these parasites. More importantly, while the study provides additional phenotypic analysis of this mutant, the data do not yet support the author’s main conclusion that all of the changes are due to defects in GPI biosynthesis and GPI-protein assembly/trafficking. While the authors provide evidence that loss of GlcNAc6P-AT leads to a global defect in GPI, the study does not address the potential role of free versus protein-linked GPIs, or whether the concurrent loss of dolichol-oligosaccharides and/or protein N-glycosylation could also contribute to the phenotypes and parasite death. Further studies would therefore be needed to support the authors key conclusions and to progress the findings beyond what has been previously reported.

**Part II – Major Issues: Key Experiments Required for Acceptance**

Reviewer #1: (No Response)

Reviewer #2: No absolutely necessary additional experiments.

Reviewer #3: Major:

1, Throughout the paper, the authors refer to the first experimental cycle as cycle 0, which is somewhat misleading. It should be cycle 1 as it indicates the first cycle. The authors need to adjust all the cycle numbers accordingly to avoid any confusion (although the authors’ prior publications used cycle 0 to indicate cycle 1).

2, In addition to TEM, the egress defects should be better illustrated using live cell imaging. If this technique is not immediately available, the authors need to discuss this and indicate the limitations of TEM such as fixation artifacts and etc.

3, The idea that the inducible knockout (iKO) parasite was undergoing apoptosis has not been fully supported by the data. Nuclear condensation was not evident in Fig 5G/J. Other features of ‘apoptosis’ were not investigated.

4, One of the most striking features of the iKO was shown in Fig5, but it was not fully discussed by the authors. iKO seems to have more ribosomes, a phenomenon not just derived from irregular distribution. The authors focused on GPI and GPI-anchored proteins (MSP1) in this study. They should also check the literature to see whether anything is known about hexosamine biosynthesis and ribosome biogenesis.

The citation #40 was not properly used here as the study showed ribosome reduction after drug treatment, which is a totally different phenomenon than ribosome increasement upon knockout of GNA1.

Reviewer #4: (No Response)

Reviewer #5: Fig 1F. The signal for the HA-tagged protein in the immunoblot is very low/indistinct and it is hard to assess whether HA-protein is still expressed in the rapamycin-treated parasites (in which there is no background signal). Importantly, the Commassie Blue stained gels - which are used as a loading control but appear to be from different blots - suggest that less protein was loaded of the rapamycin-treated parasites. Overall, these analyses leave open the possibility that residual GlcNAc6P-AT activity may persist over more than one cycle. This is supported by the LC-MS data shown in Fig 2B-C which showed significant levels of UDP-GlcNAc and GlcNAc1/6P in the rapamycin-treated parasites. Further evidence demonstrating loss of GlcNAc6P-AT activity following knock-down and the kinetics of enzyme loss would assist with the interpretation of the other phenotypic analyses.

The authors conclude that parasite death (and all other preceding phenotypic changes) primarily reflect loss of GPI-protein expression/trafficking, although direct evidence for this conclusion is not provided. In Fig 3A, the authors show that total GPI levels have decreased in rapamycin-treated parasites - however, it is not clear when parasites were harvested (cycle, time point) and to what extent loss of GPIs tracks loss of GlcNAc6P-AT activity. The GC-MS assay also does not distinguish between free GPIs and protein-linked GPIs. Further dissection of dynamics of GPI loss versus loss of UDP-GlcNAc synthesis and parasite death are needed to support the authors conclusions.

The authors suggest that the phenotypes observed following rapamycin-treatment are not due to loss of N-glycosylation, as these effects would only manifest in the second or later cycles. However, I am not aware that this has been demonstrated definitively in the literature and also that it is likely to be very dependent on the mutant studied (for example defects in isoprenoid synthesis have pleiotropic effects). It is also worth noting that disruption of the function of just one protein due to a partial loss of N-glycosylation may be sufficient to cause parasite death. As a result, the authors really need to confirm that N-glycosylation is not affected in the first cycle and that hypo-N-glycolsylation of one or more proteins does not underlie the observed phenotypes.

Previous studies have shown that knock-down of GlcNAc6P-AT can be biochemically by-passed by supplementing cultures with GlcNAc. Inclusion of this control would provide strong support for the function of GlcNAc6PAT being primarily/exclusively enzymatic, rather than non-enzymatic, and should be included.

**Part III – Minor Issues: Editorial and Data Presentation Modifications**

Reviewer #1: I suggest adding that disrupted parasite are not able to reinvade new RBC to abstract.

Quantification of disrupted or damaged food vacuole seen in 5H would be another killing mechanism important to quantify. You might be able to score existing EM images for rough estimate of disrupted FV like 1% or 10% or more. EM only looks at a single plane and will underestimate. Might a DV membrane protein marker like PfCRT or mdr1 be used to look at the ration of intact DVs. Poosibly a acridine orange stain or another pH marker could quantify percent of ruptured DV with disruption of this pathway.

Line 379 I suggest deleting leading to apoptosis as you have not shown this mechanism. You have other mechanisms of killing. This unsubstantiated mechanism distracts from an substantial body of work.

Minor

Line 84 suggest “some previous work reports”

Line 110-please report “may” contribute. The data is weak compared to other pathogen GPI on an inflammatory response. Malaria parasite number 40 million per mL for a one percent parasitemia and the Plasmodium GPI moiety have some but minor contribution to inflammation.

Reviewer #2: This reviewer suggests that the GPI biosynthetic pathway be presented in a similarly compact fashion to that used for the UDP-GlNAc biosynthetic pathway in Fig. 1A.

This would be helpful for readers outside the GPI domain.

The authors show by imaging techniques that the distribution of GPI-anchored proteins is abnormal compared to normal conditions. In my opinion, further clarification is needed. Is it caused by structural deviations of the GPI parts of the proteins? Are they proteins before transmission of the anchor that are anchored via their non cleaved hydrophobic end in the membrane? This could be clarified in part with the help of phospholipases or glycosidases.

In any case, the authors should at least include these aspects in their discussion.

Reviewer #3: Minor:

1, Fig 1A. Please illustrate the structures of the molecules in the HBP pathway.

2, Fig 1E. Please clearly describe how invasion rate was calculated. Was it the ratio of ring% compared to schizonts% of the prior cycle?

3, Fig 1F. Please comment on the protein expression level of GNA1. How much protein was loaded? The band ~ 30 kDa is barely visible. There is a more abundant band around 10 kDa. Was it a degradation product or a partially cleaved product of GNA1?

4, Line 126, the correct citation should be #5, not #6.

5, Line 233. The authors should indicate how many parasites were investigated and what is the percentage of normal MSP1 distribution in the inducible KO.

6, Fig 3C. In blots with MSP1 and AMA1, please indicate which band is the full-length protein and what are the bands of cleaved (or processed) products.

7, Line 246. This statement should be supported by quantification of band intensities. Fig 3C showed the bands in the Triton/Rapamycin lane were fainter compared to Triton/DMSO. It remains unclear whether MSP1 processing was affected.

8, Line 280. This citation was not numbered.

9, Line 323. The authors need to indicate how many parasites were investigated and what is the percentage of normal segmented parasites in the inducible KO.

10, Line 379. Apoptosis was not fully verified in this study.

11, Supplementary Typos, Line 54, y F4

12, Fig S3A, please indicate the positions of the start and stop codons.

Reviewer #4: (No Response)

Reviewer #5: Do the authors have any thoughts on why other metabolites unrelated to hexosamine synthesis (e.g N-carbamoyl-Asp) are elevated in the knock-out?

PLOS authors have the option to publish the peer review history of their article (what does this mean? ). If published, this will include your full peer review and any attached files.

**Do you want your identity to be public for this peer review?** For information about this choice, including consent withdrawal, please see our Privacy Policy .

Reviewer #1: No

Reviewer #2: No

Reviewer #3: No

Reviewer #4: No

Reviewer #5: No

**Figure resubmission:**

**Reproducibility:**



---

## [Decision Letter · Decision Letter 1]

2 Jun 2025

Dear Dr. Izquierdo,

We are pleased to inform you that your manuscript 'Hexosamine Biosynthesis Disruption Impairs GPI Production and Arrests Plasmodium falciparum Growth at Schizont Stages' has been provisionally accepted for publication in PLOS Pathogens.

Best regards,

Laura Kirkman, MD

Academic Editor

PLOS Pathogens

Tracey Lamb

Section Editor

PLOS Pathogens

Sumita Bhaduri-McIntosh

Editor-in-Chief

PLOS Pathogens

orcid.org/0000-0003-2946-9497

Michael Malim

Editor-in-Chief

PLOS Pathogens

orcid.org/0000-0002-7699-2064

Reviewer Comments (if any, and for reference):

Reviewer's Responses to Questions

**Part I - Summary**

Reviewer #4: The authors have responded productively to each of the reviewers.

Reviewer #5: This study investigates the consequences of disrupting hexosamine biosynthesis (specifically

GlcNAc6P synthesis) in the malaria parasite, Plasmodium falciparum. In this revised manuscript, the authors have addressed several of the major issues raised in the first review. Specifically, they present new data which support the conclusion that the phenotypes observed following disruption of hexosamine biosynthesis are largely due to a block in GPI biosynthesis rather than N-glycosylation. The use of exogenous GlcNAc to rescue lines until hexosamine synthesis had been inhibited in rapamycin-treated parasites was a clever addition. I thank the authors for their efforts in addressing the comments of all reviewers.

**Part II – Major Issues: Key Experiments Required for Acceptance**

Reviewer #4: I have no further comments

Reviewer #5: None

**Part III – Minor Issues: Editorial and Data Presentation Modifications**

Reviewer #4: None noted

Reviewer #5: (No Response)

PLOS authors have the option to publish the peer review history of their article (what does this mean? ). If published, this will include your full peer review and any attached files.

**Do you want your identity to be public for this peer review?** For information about this choice, including consent withdrawal, please see our Privacy Policy .

Reviewer #4: No

Reviewer #5: No

---

## [Editor Report · Acceptance letter]

Dear Dr Izquierdo,

We are delighted to inform you that your manuscript, "Hexosamine Biosynthesis Disruption Impairs GPI Production and Arrests Plasmodium falciparum Growth at Schizont Stages," has been formally accepted for publication in PLOS Pathogens.

Best regards,

Sumita Bhaduri-McIntosh

Editor-in-Chief

PLOS Pathogens

orcid.org/0000-0003-2946-9497

Michael Malim

Editor-in-Chief

PLOS Pathogens

orcid.org/0000-0002-7699-2064